# Histone variant H2A.J accumulates in senescent cells and promotes inflammatory gene expression

Kévin Contrepois[1,2,*], Clément Coudereau[1,*], Bérénice A. Benayoun[2,3], Nadine Schuler[4], Pierre-François Roux[5], Oliver Bischof[5], Régis Courbeyrette[1], Cyril Carvalho[1], Jean-Yves Thuret[1], Zhihai Ma[2], Céline Derbois[6], Marie-Claire Nevers[7], Hervé Volland[7], Christophe E. Redon[8], William M. Bonner[8], Jean-François Deleuze[6], Clotilde Wiel[9], David Bernard[9], Michael P. Snyder[2], Claudia E. Rübe[4], Robert Olaso[6], François Fenaille[10] & Carl Mann[1]

The senescence of mammalian cells is characterized by a proliferative arrest in response to stress and the expression of an inflammatory phenotype. Here we show that histone H2A.J, a poorly studied H2A variant found only in mammals, accumulates in human fibroblasts in senescence with persistent DNA damage. H2A.J also accumulates in mice with aging in a tissue-specific manner and in human skin. Knock-down of H2A.J inhibits the expression of inflammatory genes that contribute to the senescent-associated secretory phenotype (SASP), and over expression of H2A.J increases the expression of some of these genes in proliferating cells. H2A.J accumulation may thus promote the signalling of senescent cells to the immune system, and it may contribute to chronic inflammation and the development of aging-associated diseases.

[1] Institute for Integrative Biology of the Cell (I2BC), CEA, CNRS, Univ. Paris-Sud, Université Paris-Saclay, 91198 Gif-sur-Yvette cedex, France. [2] Department of Genetics, Stanford University, Stanford, California 94305-5120, USA. [3] Paul F. Glenn Laboratories for the Biology of Aging, Stanford University, Stanford, California 94305-5120, USA. [4] Department of Radiation Oncology, Saarland University, 66421 Homburg (Saar), Germany. [5] Institut Pasteur/INSERM U933, Laboratory of Nuclear Organization and Oncogenesis, Department of Cell Biology and Infection, 75015 Paris, France. [6] CEA, CNG, 91057 Evry, France. [7] CEA, Service de Pharmacologie et Immunoanalyse (SPI), INRA, Université Paris-Saclay, F-91191 Gif-sur-Yvette, France. [8] Laboratory of Molecular Pharmacology, C.C.R., N.C.I., N.I.H., Bethesda, Maryland 20892, USA. [9] Inserm U1052, Centre de Recherche en Cancérologie de Lyon, CNRS UMR5286, Centre Léon Bérard, Université de Lyon 69008, Lyon, France. [10] CEA, IBITECS, Service de Pharmacologie et d'Immunoanalyse, UMR 0496, Laboratoire d'Etude du Métabolisme des Médicaments, MetaboHUB-Paris, Université Paris Saclay, F-91191 Gif-sur-Yvette cedex, France. * These authors contributed equally to this work. Correspondence and requests for materials should be addressed to C.M. (email: carl.mann@cea.fr).

Mammalian cellular senescence is a process in which cells lose their ability to proliferate, accompanied in most cases by the expression of an inflammatory phenotype called the senescent-associated secretory phenotype (SASP)[1]. Cellular senescence has most often been studied as a response to stresses that can damage DNA or destabilize the genome, such as the loss of telomere sequences or oxidative stress. Remarkably, senescence can also be induced by the expression of hyper-mitogenic oncogenes in non-transformed cells[2]. These features led to the recognition of senescence as an important tumour suppressor mechanism that blocks the proliferation of cells with tumorigenic potential. The SASP has been implicated in the signalling of senescent cells to the immune system for their elimination and for wound healing[1,3–5]. Recent data suggest that there are functionally distinct senescent states depending on the stress-inducing condition, the cell type, and the time that the cells were maintained in senescence[6]. Important distinctions include senescence with or without persistent DNA damage that would lead to the activation of distinct signalling pathways. Unfortunately, few molecular correlates and biomarkers have been defined for these senescent states. The chromatin of senescent cells is a promising area to explore because senescent cells have striking modifications in chromatin that likely contribute to differential genome expression and the maintenance of the senescent state[7,8]. Chromatin is composed of DNA wrapped around nucleosomes that are formed from histones and associated proteins that bind DNA or the histones. The canonical histones are highly synthesized in S phase to package the newly replicated DNA[9]. Non-canonical histone variants are endowed with specific functional properties determined by their diverged protein sequences and their constitutive expression in contrast to the replication-dependent expression of the canonical histones[10]. Some variants are highly diverged, whereas others, such as H3.3, exhibit major functional differences with just four amino acid substitutions relative to canonical H3.2 (ref. 11). Recent examples of roles for histone variants in senescence include an N-terminal proteolysis of histone H3.3 in senescence that was implicated in the repression of proliferation genes[12], and a role for macro-H2A1 in the expression and the feedback regulation of SASP gene expression during RASval12-induced senescence[13]. The histone H3-K4 methyl-transferase MLL1 was also shown to be indirectly required for expression of the SASP during oncogene-induced senescence through the transcriptional activation of pro-proliferative genes and activation of the ATM kinase[14].

In this work, we describe the first, to the best of our knowledge, characterization of histone variant H2A.J, that differs from canonical H2A by only five amino acids, and its putative functional importance in senescence, aging and cancer.

## Results

**H2A.J accumulates in senescent fibroblasts with DNA damage.** We used mass spectrometry to analyze histones in human fibroblasts in proliferation, quiescence (serum starvation), and various senescent states using a combined top-down and bottom-up approach that we developed[15,16]. As previously described[16], we examined fibroblasts in replicative senescence, oncogene-induced senescence, and DNA-damage-induced senescence. We also compared cells maintained in senescence or quiescence for short (5 days, early) or longer (20 days, deep) time periods (Fig. 1a). Replicative senescence of non-immortalized fibroblasts was induced by the continual passage of cells until the proliferative arrest of the cultures (∼65 population doublings). Oncogene-induced senescence was provoked by the expression of activated forms of the RAF1 kinase or by RASval12 in WI-38 or IMR90 fibroblasts

immortalized with hTERT, and sustained exposure to 20 μM etoposide was used to induce senescence of WI-38hTERT fibroblasts by the creation of persistent DNA double-strand breaks. Senescence was verified by the induction of a durable proliferative arrest, the expression of senescence-associated β-galactosidase activity (SA-β-gal), the cell cycle inhibitors p16 and p21, and a characteristic senescence transcriptome (see below).

Remarkably, we found that the H2A.J histone variant (Uniprot Q9BTM1, encoded by the unique *H2AFJ* gene) is present at low levels (≈1% of canonical H2A species) in proliferating human fibroblasts, but accumulated ≈10-fold in the chromatin of fibroblasts in senescence with persistent DNA damage (replicative senescence and etoposide-induced senescence) over a period of 20 days (Fig. 1b–e and Supplementary Figs 1 and 2). Our histone profiling also revealed changes in the levels of some canonical H2A species. The proportion of canonical H2A-type 1C (Uniprot Q93077) increased in all non-proliferative states (quiescence and senescence), whereas other canonical histones, such as the H2A-type 1 (Uniprot P0C0S8) and H2A-type 1B/E (Uniprot P04908), decreased in senescent cells with persistent DNA damage. However, H2A.J was the only H2A species that was present at low levels in proliferating cells and high levels in senescent cells with DNA damage.

The human genome contains 26 genes encoding 11 canonical H2A species and 8 H2A variants (Supplementary Fig. 2 and http://www.actrec.gov.in/histome/histones.php?histone=H2A)[17]. H2A.J is a poorly studied H2A variant that has never been described at the protein level. It differs from canonical H2A protein sequences only by an A11V substitution and the presence of an SQK motif near the C-terminus (Fig. 1d and Supplementary Fig. 2). Phylogenetic analyses indicate that H2A.J is specific to mammalian organisms (Supplementary Fig. 3). The canonical histone genes are principally expressed in S phase, whereas the *H2AFJ* gene is expressed constitutively through the cell cycle[18]. Canonical histones are encoded by multiple genes that are mostly clustered on human chromosomes 1 and 6, whereas the unique *H2AFJ* gene is located on human chromosome 12. These features of the H2A.J histone variant and its striking accumulation in senescent fibroblasts with persistent DNA damage incited us to embark on its detailed characterization.

In contrast to replicative and etoposide-induced senescence, fibroblasts in quiescence or RAF-induced senescence contained few cells with persistent γH2AX foci (Fig. 1e) and contained lower levels of H2A.J (Fig. 1b,c). These results were confirmed by Western blotting (Fig. 1f) with a homemade anti-H2A.J antibody (see below and Methods section for a description of the antibody). Thus, maximal accumulation of H2A.J was correlated with a senescent proliferative arrest associated with persistent DNA damage. A detailed time course tracking H2A.J accumulation in WI-38hTERT cells induced into senescence with etoposide showed that H2A.J levels increased rapidly for 1–2 weeks and then more slowly over the next month (Fig. 1g). This extended accumulation is not consistent with an acute short-term response to DNA damage.

Senescence of human fibroblasts induced by the RASval12 oncogene is associated with a higher level of DNA damage than senescence induced by an activated RAF kinase[19–21]. Consistent with this difference, we observed an accumulation of H2A.J in IMR90hTERT fibroblasts induced into senescence by RASval12 (Fig. 2a,b). We also exploited the well-known model of DMBA/TPA skin carcinogenesis[22] to test for accumulation of H2A.J in an in vivo model of RAS-induced senescence of mouse epithelial cells[23]. Treatment of mouse skin with a single application of the carcinogen 7,12-dimethylbenz[a]-anthracene (DMBA) mutagenizes the epidermal cells. Activating mutations of the *H-RAS* gene within keratinocyte stem cells are initiating events for tumorigenesis.

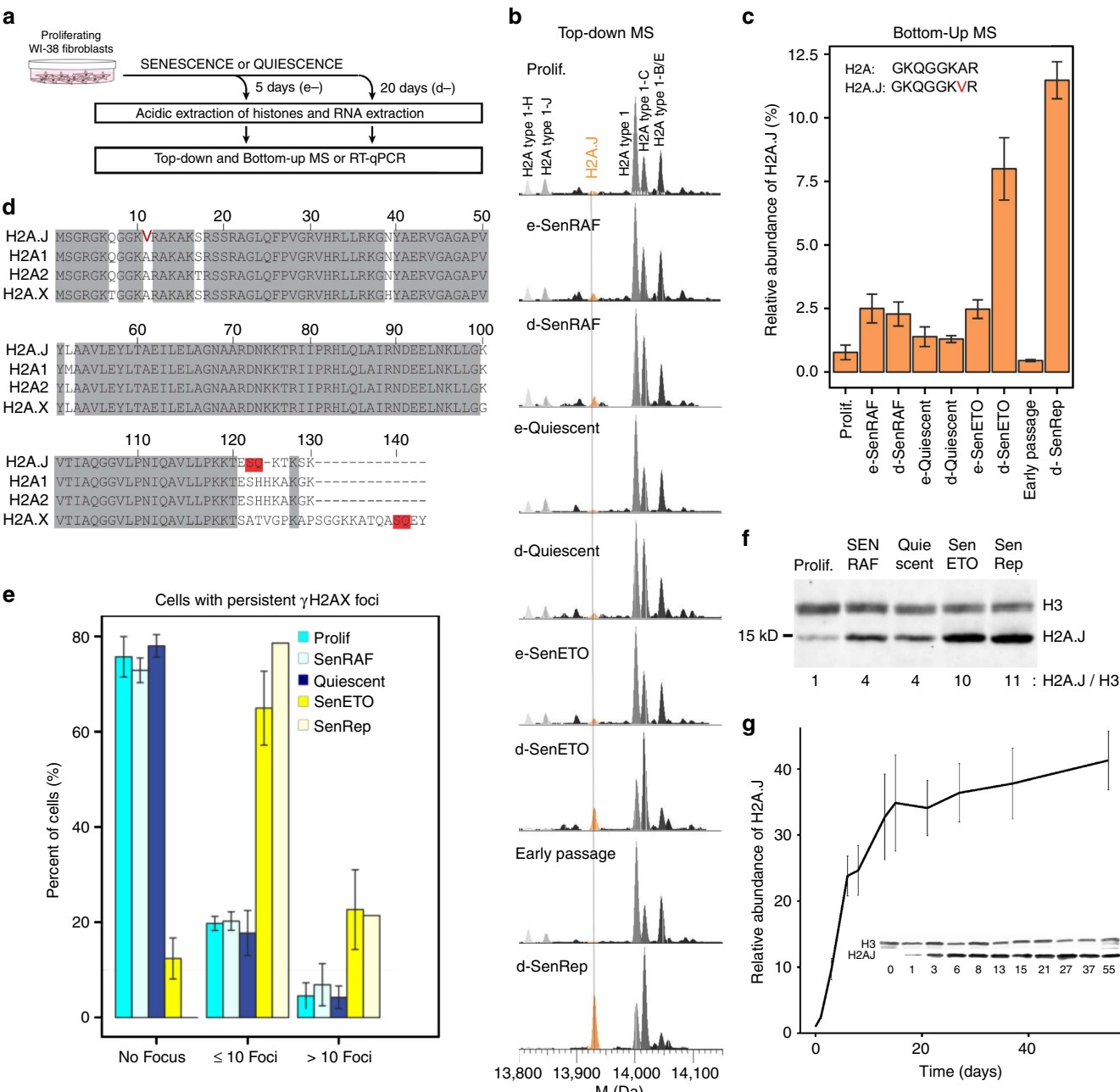

**Figure 1 | H2A.J accumulates in senescent human fibroblasts with persistent DNA damage.** (**a**) Experimental plan. (**b–f**) WI-38hTERT-GFP-RAF-ER fibroblasts[21] were induced into senescence by a hyperactive RAF kinase for 5 days (e-SenRAF) or 20 days (d-SenRAF), into quiescence by serum starvation for 5 days (e-Quiescent) or 20 days (d-Quiescent), or into senescence by treatment with 20 μM etoposide for 5 days (e-SenETO) or 20 days (d-SenETO). Histones and RNAs were extracted from these cells and from non-immortalized proliferating WI-38 fibroblasts (Early Passage) and the same cells maintained in replicative senescence for 20 days (d-SenRep). (**b**) Top-down deconvoluted mass spectra of intact histone H2A2 proteins showing accumulation of H2A.J (orange) only in samples derived from senescent cells with persistent DNA damage (d-SenETO and d-SenRep-see **e** below). Canonical H2A2 species are in grey (Supplementary Figs 1,2 for details). (**c**) Relative abundance of H2A.J versus all canonical H2A species determined by quantifying the Gly5-Arg12 tryptic peptide from H2A.J versus the same peptide from all canonical H2A species (Fig. 1d and Supplementary Fig. 2). Shown are the mean and s.d. for 4 biological replicates, except 'Early Passage' and 'd-SenRep' ( 2 biological replicates). (**d**) Sequence alignment of human H2A.J, H2A fraction 1 (H2A1), H2A fraction 2 (H2A2), and H2A.X. H2A.J is distinguished by Val11 and, like the H2A.X variant, by the presence of an SQ motif near the C-terminus (boxed in red). (**e**) Percent cells expressing the indicated number of γH2AX foci, a marker of DNA damage. Most SenETO and SenRep fibroblasts contained multiple foci, whereas most fibroblasts in proliferation, quiescence, or senescence induced by a hyperactive RAF kinase (SenRAF) contained no foci. Mean and s.d. from a minimum of 2 biological replicates. (**f**) Western blot of acid-extracted histones from fibroblasts as indicated. The H2A.J/H3 levels were normalized to the ratio in proliferation. (**g**) Time course of H2A.J accumulation during etoposide-induced senescence of WI-38hTERT fibroblasts by Western blotting. The error bars show the s.d. from three biological replicates.

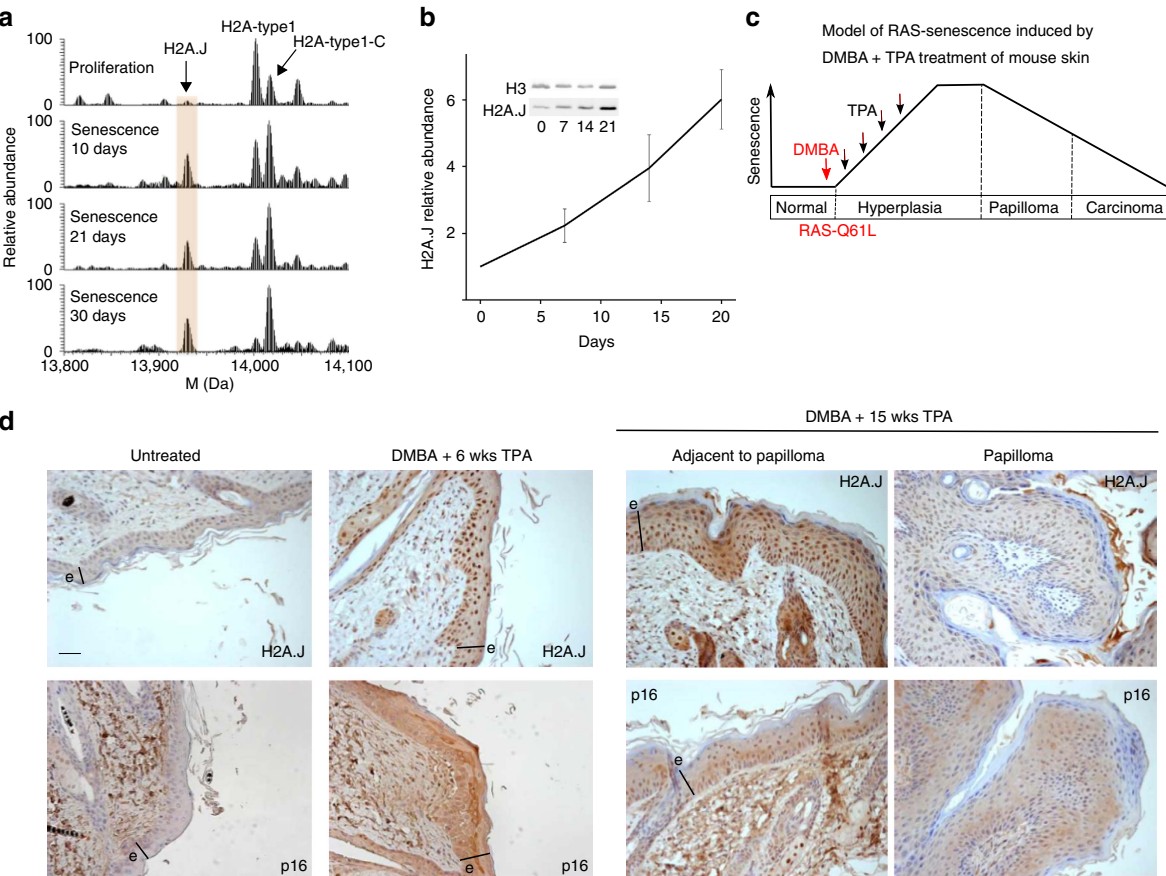

**Figure 2 | H2A.J accumulates during RAS-induced senescence *in vitro* and *in vivo*.** (**a**) H2A.J accumulates in IMR90hTERT fibroblasts induced into senescence by activation of an ER-RASval12 fusion protein with 4-hydoxy-tamoxifen (4-HT). Shown are top-down deconvoluted mass spectra of intact H2A2 proteins. The H2A.J peak increased significantly by 10 days of RAS-induced senescence. The sample of cells in senescence for 30 days represents an independent biological replicate relative to the samples in 10 and 21 days of senescence. (**b**) Immunoblot showing the time course of accumulation of H2A.J after addition of 4-HT to induce senescence. The H2A.J relative abundance was determined by normalizing the H2A.J/H3 signal to that of proliferating cells (0 days 4-HT). The error bars show the s.d. of two technical replicates. These data represent an independent biological replicate relative to the two cultures in **a**. (**c**) Schematic model showing that DMBA treatment induces the RAS-Q61L activating mutation[22] and subsequent TPA treatment stimulates proliferation and the induction of senescence markers[23]. Continued TPA treatment leads to occasional papillomas that have escaped or bypassed senescence. (**d**) H2A.J accumulates in mouse epidermal cells in a model of RAS-induced senescence *in vivo*. Immunohistochemical analysis of H2A.J and p16-INK4A in mouse sections treated once with DMBA and then twice weekly with TPA for 6 or 15 weeks. Note the increased H2A.J and p16 staining in the epidermis treated with DMBA + 6 weeks of TPA. The epidermal layer is indicated by the black line labelled 'e'. Scale bars, 30 μm. Occasional papillomas that developed after DMBA + 15 weeks of TPA treatment showed reduced H2A.J and p16 staining compared to the adjacent treated skin.

However, tumour progression requires promotion by repeated application of 12-*O*-tetradecanoylphorbol-13-acetate (TPA) that stimulates the proliferation of the keratinocytes expressing activated RAS (Fig. 2c). We previously showed that DMBA + TPA-treated skin becomes populated with senescent keratinocytes as revealed by senescent markers such as SA-β-gal, p16, DEC1 and DcR2 (ref. 23). Continued TPA treatment leads to papillomas that are composed of keratinocytes showing reduced expression of these senescent markers and increased expression of Ki-67. Immuno-histochemical staining showed that H2A.J and p16 levels increased in mouse epidermis treated with DMBA + TPA for 6 weeks (Fig. 2d). Further treatment until 15 weeks led to the outgrowth of occasional papillomas. H2A.J and p16 staining was elevated in the skin adjacent to the papillomas, but reduced in the cells forming the papilloma. These data are consistent with an accumulation of H2A.J in keratinocytes induced into senescence by the active RAS oncogene *in vivo*.

**H2A.J accumulation involves a post-transcriptional component.** We tested whether mRNA levels were correlated with the protein

levels of the H2A variants (Fig. 3 and Supplementary Fig. 4). Unlike most mRNAs, canonical histone mRNAs contain a 3′ untranslated region (UTR) stem-loop motif instead of a polyA tail. To provide histones for packaging newly synthesized DNA into nucleosomes, canonical histone mRNA levels peak in S phase through transcriptional activation and post-transcriptional processing of mRNA at the 3′-UTR stem-loop motif by the SLBP (stem-loop binding protein) pathway[9]. SLBP and canonical histone mRNAs are turned over rapidly in non-proliferating cells. With the exception of *HIST1H2AC* mRNA, most of the canonical H2A mRNAs dropped dramatically in non-proliferating cells (Fig. 3a), as expected from their known transcriptional and post-transcriptional regulation[9]. The persistence of *HIST1H2AC* mRNA in non-proliferating cells may be explained by the presence of a polyadenylation site downstream of its stem-loop motif. Polyadenylated *HIST1H2AC* mRNA indeed accumulated in quiescent and senescent fibroblasts (Fig. 3b) and likely contributes to the accumulation of H2A-type 1C protein in these states (Fig. 1b). In contrast to the canonical histone genes, *H2AFJ* expression was equal or higher in non-proliferating (quiescent and senescent) human fibroblasts compared to proliferating fibroblasts (Fig. 3a and Supplementary Fig. 4),

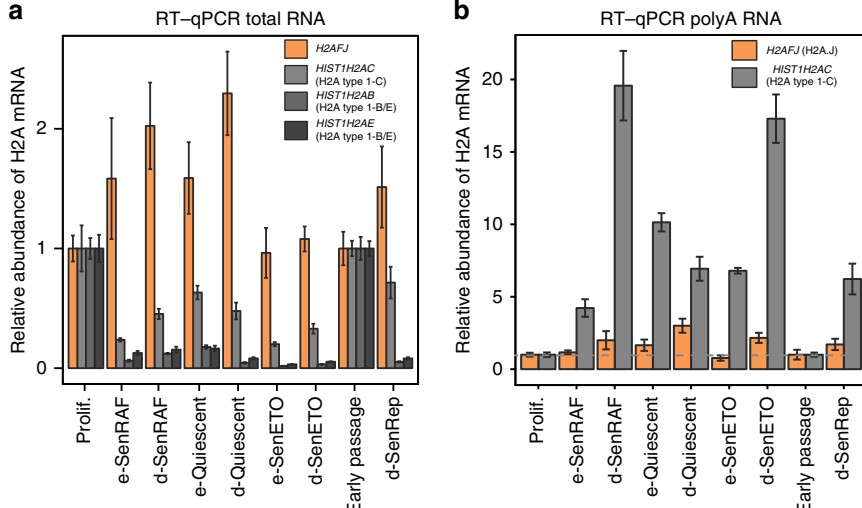

**Figure 3 | RNA levels of *H2AFJ* and selected canonical H2A species.** RNA levels are compared in senescent or quiescent cells relative to proliferating cells. Random primers were used to prepare cDNA to quantify total RNA (**a**) whereas oligo dT was used to prepare cDNA to quantify polyA RNA (**b**). Shown are the mean and s.d. for triplicate qPCR values from single samples. A second biological replicate from total RNA is shown in Supplementary Fig. 4. RNA levels were normalized to *GAPDH* mRNA and to the levels of the respective histone mRNAs in proliferating fibroblasts. RNAs were extracted from proliferating WI-38hTERT-GFP-RAF-ER fibroblasts (Prolif.), the same cells induced into quiescence by serum starvation for 5 days (e-Quiescent) or 20 days (d-Quiescent), induced into senescence by the expression of a hyperactive RAF kinase for 5 days (e-SenRAF) or 20 days (d-SenRAF), or induced into senescence by treatment with etoposide for 5 days (e-SenETO) or 20 days (d-SenETO). RNAs were also extracted from non-immortalized proliferating WI-38 fibroblasts (Early Passage) and the same cells maintained in replicative senescence for 20 days (d-SenRep).

indicating that its expression is replication-independent, as described previously[18,24]. *H2AFJ* mRNA was not higher in fibroblasts in replicative or etoposide-induced senescence compared to fibroblasts in RAF-induced senescence or quiescence. Thus, H2A.J accumulation during senescence with persistent DNA damage must involve a post-transcriptional component in addition to the continued expression of its gene in non-proliferating cells.

**H2A.J promotes inflammatory gene expression in senescence.** We used two different lentiviral shRNA constructs (sh2 and sh3) to search for phenotypes associated with the knock-down of H2A.J in fibroblasts (Supplementary Fig. 5). We also ectopically expressed an sh-resistant *H2AFJ* cDNA to test its ability to rescue the *H2AFJ* knock-down phenotypes and to study the effect of the over expression of H2A.J in proliferating cells. Depletion of *H2AFJ* mRNA blocked H2A.J accumulation in senescent cells and this effect was prevented by ectopic expression of the sh-resistant *H2AFJ* cDNA (Supplementary Fig. 5). H2A.J knock-down had no obvious effect on the proliferation of fibroblasts or the efficiency with which they entered into and maintained the senescent state after treatment with etoposide (Supplementary Fig. 6), nor did H2A.J depletion significantly impact cell morphology, DNA compaction, or the distribution of γH2AX foci in etoposide-induced senescence (Supplementary Fig. 7).

However, H2A.J depletion significantly affected the gene expression of senescent cells as determined by transcriptome analyses (Illumina Bead Chip arrays and RNA-seq) (Fig. 4a). Human fibroblasts induced into senescence by etoposide treatment for three weeks had a greatly modified transcriptome relative to proliferating cells: 3,471 genes were differentially expressed at least twofold (1,821 up-regulated and 1,650 down-regulated) with a false-discovery rate (FDR) of <0.01 (Fig. 4b and Supplementary Data 1). Down-regulated genes were greatly enriched in gene ontology (GO) terms involving cell proliferation as expected for the proliferative arrest characteristic of senescence (Supplementary Data 2). Up-

regulated genes were greatly enriched in GO terms related to inflammatory/immune/secreted proteins (35/50 top GO classes) as previously reported for senescent cells (Supplementary Data 2). The Excel files list the 50 most significant GO IDs/terms revealed by the goana function in limma[25] along with the *P* values for the GO ID enrichment and the gene symbols for the differentially-expressed genes that belong to each GO ID. Similar biological processes or molecular functions are highlighted in orange, and cellular compartment IDs are highlighted in yellow.

Only 12 genes (including *H2AFJ*) were significantly down-regulated and 20 genes up-regulated by 2 different sh-RNAs targeting *H2AFJ* in proliferating fibroblasts (Supplementary Data 3). However, H2A.J knock-down had a much greater effect in senescent cells. 165 genes were down-regulated and 83 genes up-regulated in senescent cells by H2A.J depletion mediated by 2 different shRNAs (FDR<0.05) (Fig. 4c and Supplementary Data 3). Remarkably, gene ontology analysis indicated that deregulated genes were enriched (43/50 top GO classes) in pathways involving immune processes, secreted gene products, and cell migration/chemotaxis (Supplementary Data 2). These GO terms are very similar to those corresponding to genes up-regulated in senescence. Indeed, 71/165 genes that were down-regulated in senescence after depletion of H2A.J represent genes that were up-regulated at least twofold in senescent versus proliferating cells (Fig. 4d, and blue dots in the Volcano plots of Fig. 4b,c). Thus, 43% of genes regulated positively by H2A.J are also up-regulated in senescent versus proliferating cells. We further analyzed this subset of genes whose upregulation in senescent cells is promoted by H2A.J. 29/71 (41%) of these genes encode proteins that are bound to the cell surface or secreted from cells, involved in immune response, or implicated in cellular senescence (Fig. 4d-asterisks). Notable genes in this class are involved in inflammation and immune response, including *CXCL1*, *CXCL10*, *CCL2*, *CCL13*, *IL1B*, *IL12A*, *VCAM1* and the *HLA-A/B/C* major histocompatibility class I genes.

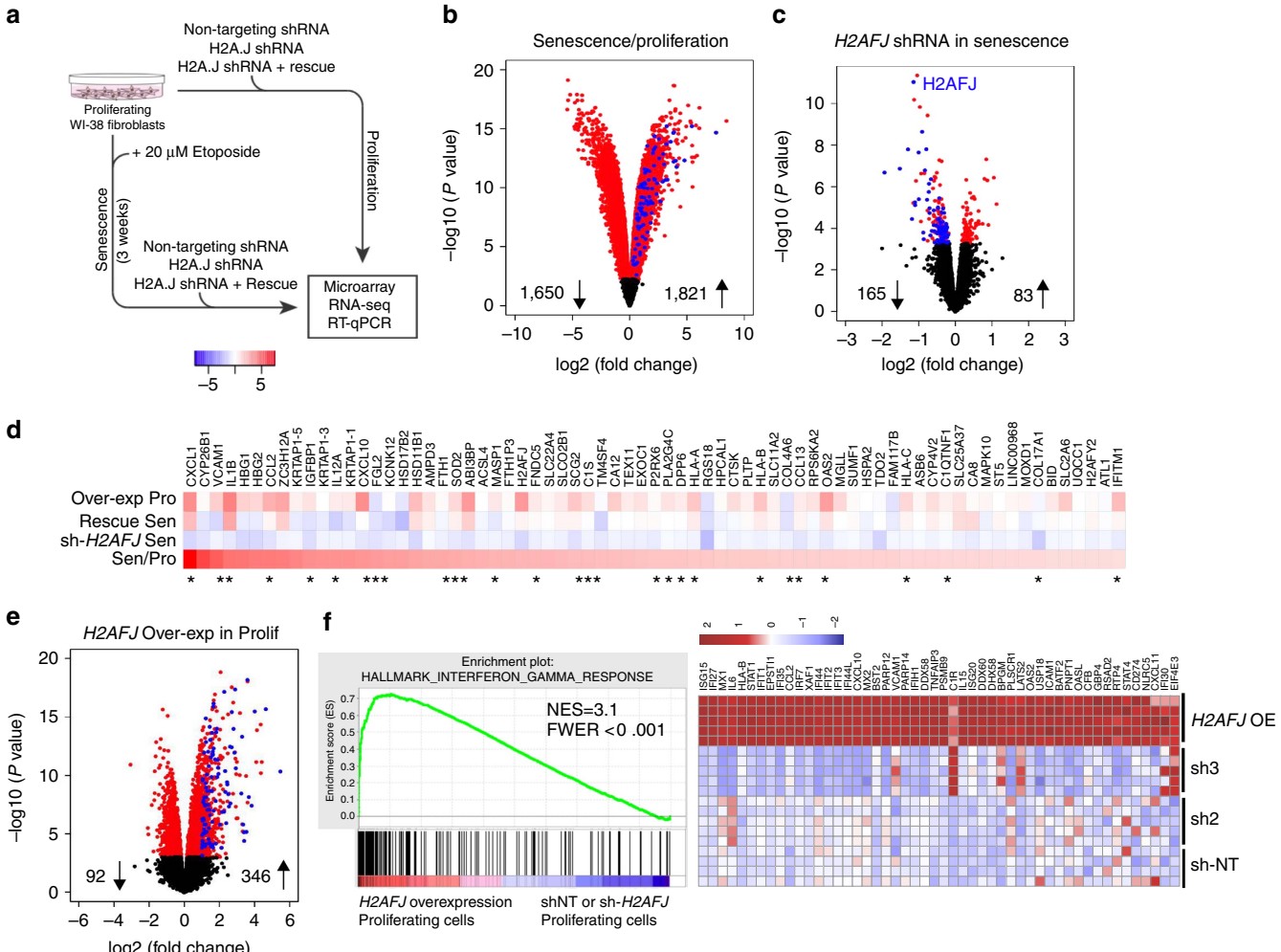

**Figure 4 | H2A.J promotes expression of immune response and inflammatory genes.** (**a**) Experimental schema of samples for microarray, RNA-seq, and RT-qPCR analyses. (**b,c**) Volcano plots summarizing the differentially expressed (DE) genes for WI-38hTERT fibroblasts induced into senescence with etoposide for 3 weeks versus proliferating fibroblasts. (**b**) Red dots: FDR < 0.01. Blue dots: genes that are transcriptionally up-regulated in senescence and down-regulated by sh-*H2AFJ* (see below). (**c**) DE genes for senescent WI-38hTERT fibroblasts expressing sh-*H2AFJ* RNAs versus a control sh-NoTarget. Only genes that were affected by both the sh2-*H2AFJ* and sh3-*H2AFJ* RNAs are shown. Red dots: FDR < 0.05. Blue dots: genes that are up-regulated in senescence in sh-NT cells. (**d**) Heat maps of log2 fold change for 71 genes that are up-regulated at least 2 fold in senescent versus proliferating fibroblasts (Sen/Pro) and down-regulated in senescence by knock-down of H2A.J (sh-*H2AFJ* Sen). These 71 genes are represented as blue dots in **c**. Also shown are the effects of ectopic expression of an sh2-resistant *H2AFJ* cDNA in senescent cells in which endogenous *H2AFJ* RNA is knocked-down with sh2 RNA (Rescue Sen), and the effect of ectopic expression of the sh-resistant *H2AFJ* cDNA in proliferating fibroblasts (Over-exp Pro). The asterisks indicate genes encoding proteins that are bound to the cell surface or secreted from cells, involved in immune response, or implicated in cellular senescence. (**e**) Volcano plot of DE genes in proliferating cells over-expressing *H2AFJ*. The blue dots show 110/346 up-regulated 'immune system process' genes belonging to GO:0002376 (Supplementary Data 2). (**f**) GSEA enrichment plot (left panel) showing the ranking of interferon gamma response genes in the set of differentially expressed genes on over expression of *H2AFJ* in proliferating fibroblasts. Right panel: the corresponding heat map of differential expression for the top 50 IF-gamma-response genes is shown for proliferating cells over-expressing *H2AFJ* (*H2AFJ* OE) versus proliferating cells expressing sh-NoTarget or sh-*H2AFJ* RNAs (sh2 or sh3). Each row shows an independent biological replicate. NES, normalized enrichment score. FWER, family-wise error rate.

**H2A.J overexpression increases inflammatory gene expression.** We also examined the effect of ectopically expressing an sh2-resistant *H2AFJ* cDNA in proliferating and senescent cells in which the endogenous *H2AFJ* mRNA was depleted. This resulted in overexpression of H2A.J in these cells (Supplementary Fig. 5A,B). H2A.J over expression did not overtly affect cell proliferation, nor did it inhibit entry into senescence in response to etoposide treatment (Supplementary Fig. 6A,B) but it increased the expression in both senescent and proliferating cells of many genes whose expression was dependent on H2A.J in senescent cells (Fig. 4d). Over expression of H2A.J had striking effects on the overall transcriptome of fibroblasts in proliferation (346 genes up-regulated at least twofold and 92 genes down-regulated at least

twofold; Fig. 4e) and in etoposide-induced senescence (120 genes up-regulated at least twofold and 59 genes down-regulated at least twofold; Supplementary Data 4). Remarkably, topGO (Supplementary Data 2) and GSEA (Gene Set Enrichment Analysis) indicated that ectopic expression of H2A.J in proliferating cells sufficed to increase the expression of inflammatory, immune, and anti-viral genes (Fig. 4e,f) including *IL1A, IL1B, IL6, CXCL1/2/10, CCL2, CCL20, IRF7* and several interferon-inducible genes (Fig. 4d–f and Supplementary Data 4). The interferon-gamma response gene set (Fig. 4f) had the highest normalized enrichment score (NES = 3.1) amongst the Hallmark gene sets of the Molecular Signatures Database[26], followed closely by the interferon-alpha response, TNF-alpha signalling via NF-kB, and

inflammatory response gene sets (Supplementary Data 4). Since chronic inflammation is tumorigenic[27], this effect of H2A.J in promoting inflammatory gene expression may explain why the *H2AFJ* gene is amplified and over-expressed in some human cancers[28,29].

**H2A.J is required for the production of SASP factors**. We noticed that several genes down-regulated by H2A.J knock down in senescent cells (Fig. 4d) encode inflammatory cytokines/chemokines belonging to the family of senescent-associated secretory proteins (SASP). The SASP is composed of cytokines, chemokines, metallo-proteases, and growth factors that are highly expressed in some senescent cells. The Campisi lab proposed a list of 40 factors whose secretion is increased twofold or more in some senescent cells[30]. 38 of these were represented as good quality probes in the Illumina Bead Chip arrays. GSEA showed that this SASP gene set is significantly down-regulated by depletion of H2A.J in senescent cells (Fig. 5a,b). These results were confirmed by real-time quantitative PCR (RT–qPCR) analyses of selected SASP genes: *CXCL1/2/3/5/6* and *IL1B* (Fig. 5c,d). The sh3-*H2AFJ* RNA had a generally stronger effect on SASP gene expression than the sh2 RNA that may be due to a weaker knock down of *H2AFJ* by sh2 compared to sh3 (Supplementary Fig. 5A,B). Time course experiments indicated that *CXCL1/5/6* genes were slowly derepressed over a 2-week period in the senescent cells, and this derepression was strongly delayed when H2A.J accumulation was blocked with either sh2 or sh3 (Fig. 5d).

The Illumina Bead Chip transcriptome results were further verified with an independent set of RNA-seq data involving three replicates and only the sh2-*H2AFJ* RNA. The smaller sample size and the lower *H2AFJ* RNA knock down with sh2 compared to sh3 limited the number of genes that were detected as being significantly differentially affected (32 down-regulated and 22 up-regulated). 26/54 (48%) of the DE genes in the RNA-seq analysis were also identified as being differentially expressed in the Bead Chip analysis. Notably, the *CXCL1/2/5/6* genes were all identified by RNA-seq as being down-regulated in senescent cells expressing sh2-*H2AFJ* RNA, in agreement with the Bead Chip analysis (Fig. 5e and Supplementary Data 5).

The transcriptome results indicated that knock-down of H2A.J in senescent cells led to a significant decrease in the abundance of RNAs coding for the SASP. To determine whether this defect led to reduced cytokine production, we used the Luminex technology to immunoassay in parallel 63 human cytokines, chemokines, or growth/adhesion factors in conditioned media from senescent fibroblasts expressing sh3-*H2AFJ* RNA or sh-NoTarget RNA. 52 factors were secreted at levels above the background signal from the control medium, and 30 of these showed statistically significant concentration decreases in the conditioned media of cells expressing sh3-*H2AFJ* compared to the sh-NT control (Fig. 5f and Supplementary Fig. 8). Highly expressed secreted factors whose concentrations were greatly decreased by H2A.J knock down included GROa(CXCL1), ENA78(CXCL5), IP10(CXCL10) and GM-CSF(CSF2) (Supplementary Fig. 8). These results confirm an important role for H2A.J in senescence in promoting the production of inflammatory chemokines and cytokines.

**The H2A.J C-terminus is functionally essential**. We used complementation analyses to test the functional importance of the unique H2A.J C-terminus. Stable WI-38hTERT cell lines were constructed expressing the *H2AFJ*-sh3 RNA complemented by an shRNA-resistant cDNA encoding wild-type H2A.J (WT-H2A.J) or a mutant in which the C terminus specific to H2A.J was exchanged for the C terminus of the canonical H2A type 1 (H2A.J-Cter-H2A). The resulting mutant differs from canonical

H2As only by an Ala11Val substitution that is specific to H2A.J (Fig. 6a and Supplementary Fig. 2A). Control cell lines expressed only a No Target shRNA or the *H2AFJ*-sh3 RNA. *CXCL1*, *CXCL5*, *CCL2* and *IL1A* RNAs were quantified in proliferating cells and in cells induced into senescence by etoposide treatment. All of these genes were induced in senescence in the sh-No Target control cells, and their expression in senescence was inhibited by the knock-down of H2A.J and restored by the ectopic expression of WT-H2A.J. (Fig. 6b). In striking contrast, the ectopic expression of the H2A.J-Cter-H2A mutant was unable to restore inflammatory gene expression in cells in which endogenous H2A.J was depleted (Fig. 6b) despite being expressed at levels slightly higher than WT-H2A.J (Supplementary Fig. 9). Thus, the specific C-terminus of H2A.J is important for its function in promoting inflammatory gene expression in senescence.

**H2A.J chromatin deposition is increased in senescent cells**. We performed targeted chromatin immunoprecipitation (ChIP) experiments to assay the occupancy of endogenous H2A.J at the promoter and the coding regions of genes whose expression depends on H2A.J (*CCL2*, *CXCL5* and *IL1B*) and genes whose expression is independent of H2A.J (*GAPDH*, *BCL2A1*) in proliferating and senescent cells (Fig. 7). The selected *CCL2*, *CXCL5* and *IL1B* promoter regions contain NF-κB-binding motifs. The specificity of the ChIP signal was demonstrated by its strong decrease when immunoprecipitating chromatin from cells in which H2A.J was knocked down. The H2A.J ChIP signal increased 2- to 10-fold in senescent versus proliferating cells depending on the specific site tested. However, we did not observe any significant differences in occupancy of H2A.J in promoter versus coding regions or in genes that were regulated by H2A.J versus genes that were not regulated by H2A.J. These results suggest that increased levels of H2A.J in senescence lead to increased chromatin deposition of H2A.J at many genomic sites, but gene expression at only a subset of these sites is affected by H2A.J.

We used ChIP-seq to further characterize the genome-wide distribution of endogenous H2A.J compared to canonical H2A in senescent fibroblasts. Cross-correlation analysis[31] showed that the sonicated chromatin immunoprecipitated with the anti-H2A.J antibody was readily distinguished from the input chromatin (Supplementary Fig. 10A,B). In particular, the cross-correlation peak distance (178 bp) was slightly larger than that of input chromatin (160 bp) and additional peaks of the size expected for di-,tri- and tetra-nucleosomes were visible. This observation indicates that the immunoprecipitation was successful and suggests that H2A.J differentially protected cross-linked chromatin from sonication relative to bulk chromatin containing canonical H2A. However, we did not observe sharp peaks of H2A.J deposition at discrete genomic locations. The density distribution of sequenced fragments rather indicated that H2A.J is widely deposited in the chromatin of senescent cells, similarly to canonical H2A (Supplementary Fig. 10C). As for the targeted ChIP experiments (Fig. 7), the H2A.J density around genes whose expression is dependent on H2A.J in senescence (*CXCL1/5/6, IL1B*) was indistinguishable from the H2A.J density around unexpressed genes (*PF4V1*) or genes whose expression was independent of H2A.J (*GAPDH*). A meta-gene analysis further revealed that H2A.J was steeply depleted close to transcription start sites (TSS) of genes as a function of their expression level (Supplementary Fig. 10D). Canonical H2A was also depleted at TSS, and in a somewhat broader fashion within the transcribed sequences of highly expressed genes (Supplementary Fig. 10E). Depletion of H2A.J and H2A at TSS may reflect a general depletion of nucleosomes at the TSS of

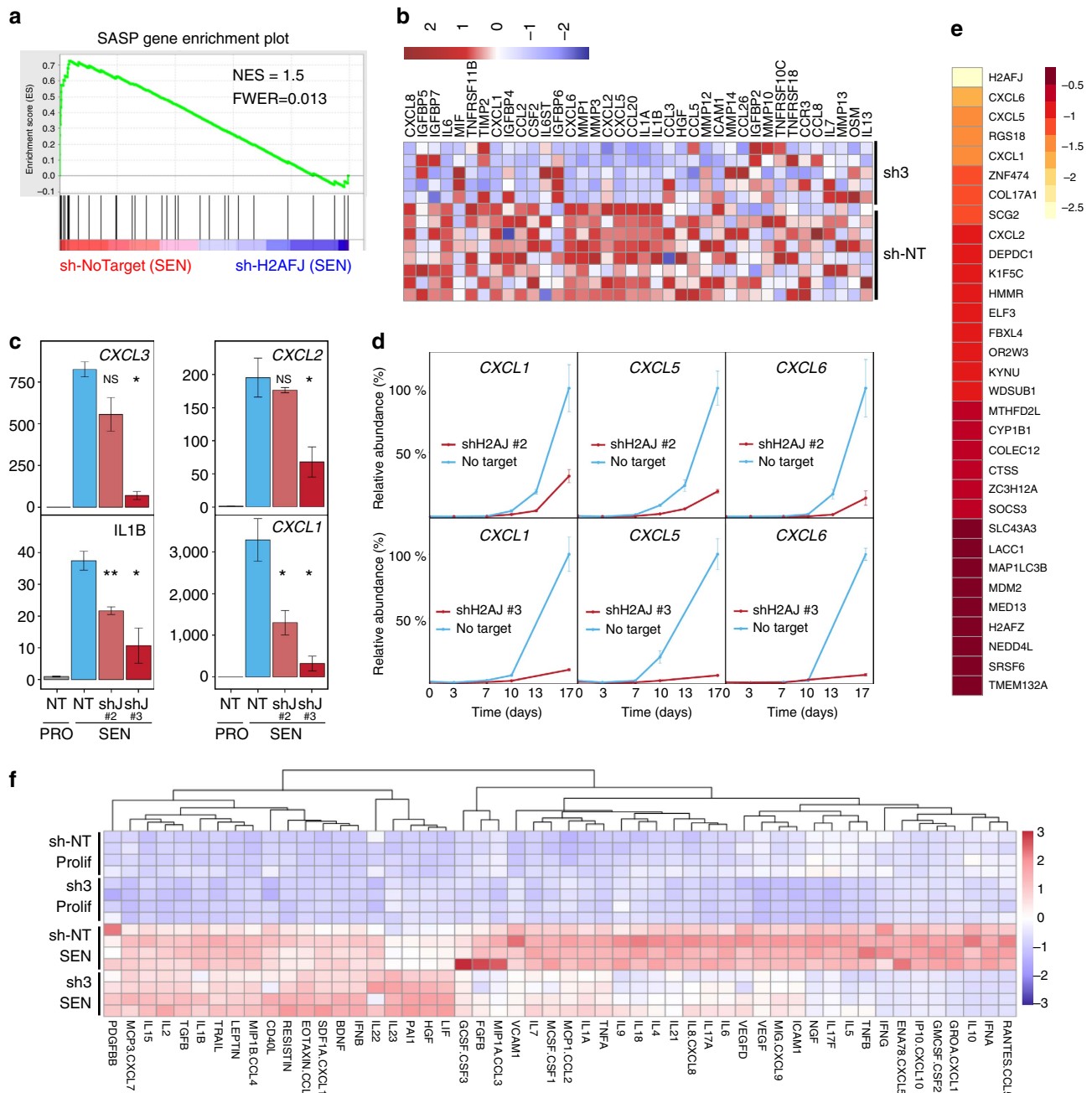

**Figure 5 | H2A.J is required for normal expression of SASP genes.** (**a**) GSEA enrichment plot showing the ranking of 38 SASP genes within the set of genes differentially-expressed on knock-down of H2A.J in senescent WI-38hTERT fibroblasts. (**b**) Corresponding heat maps for the 38 SASP genes. Each row represents an independent biological replicate of cells expressing sh3-*H2AFJ* or sh-NoTarget. Genes are listed by highest (left) to lowest (right) expression in the sh-NoTarget cells. (**c**) RT-qPCR verification of the effect of H2A.J knock-down on *CXCL1*, *CXCL2*, *CXCL3*, and *IL1B* expression in senescent cells (SEN). Values were normalized to proliferating fibroblasts expressing an sh-No Target RNA (PRO-NT). Shown are 3 biological replicates for senescent cells expressing No Target or sh-*H2AFJ* RNAs. Error bars represent mean ± s.d. *t*-tests with *$P < 0.05$, **$P < 0.001$ or not significant (NS). (**d**) Time course of *CXCL1*, *CXCL5*, *CXCL6* derepression during etoposide-induced senescence for cells expressing an sh-NoTarget or the indicated sh-*H2AFJ* RNA. The maximal expression for each gene as determined by qPCR was normalized to 100% (3 technical replicates). (**e**) Heat map showing genes significantly down-regulated in senescence after knock-down of *H2AFJ* by sh2-RNA as determined by RNA-sequencing. (**f**) Heat map constructed with the pheatmap R package showing the scaled concentrations of 52 human cytokines, chemokines, growth or adhesion factors in the conditioned media of WI-38hTERT fibroblasts expressing sh-NT or sh3-*H2AFJ* RNAs in proliferation or senescence. See Supplementary Fig. 8 for dot plots showing concentrations (pg ml$^{-1}$) of secreted factors that are differentially affected by H2A.J knock-down.

transcribed genes, and competition with deposition of the H2A.Z variant that is enriched at TSS[32]. This genome-wide analysis reinforces our conclusion that the specificity of H2A.J action is not explained by its differential deposition at the genes that it regulates.

**H2A.J is a biomarker of some senescent stem and skin cells.** The accumulation of H2A.J in senescent cells with persistent DNA damage suggested that it might be a novel biomarker for this state. We tested this possibility in the mouse. The median lifespan of C57BL/6 mice is a little more than 2 years[33]. We first

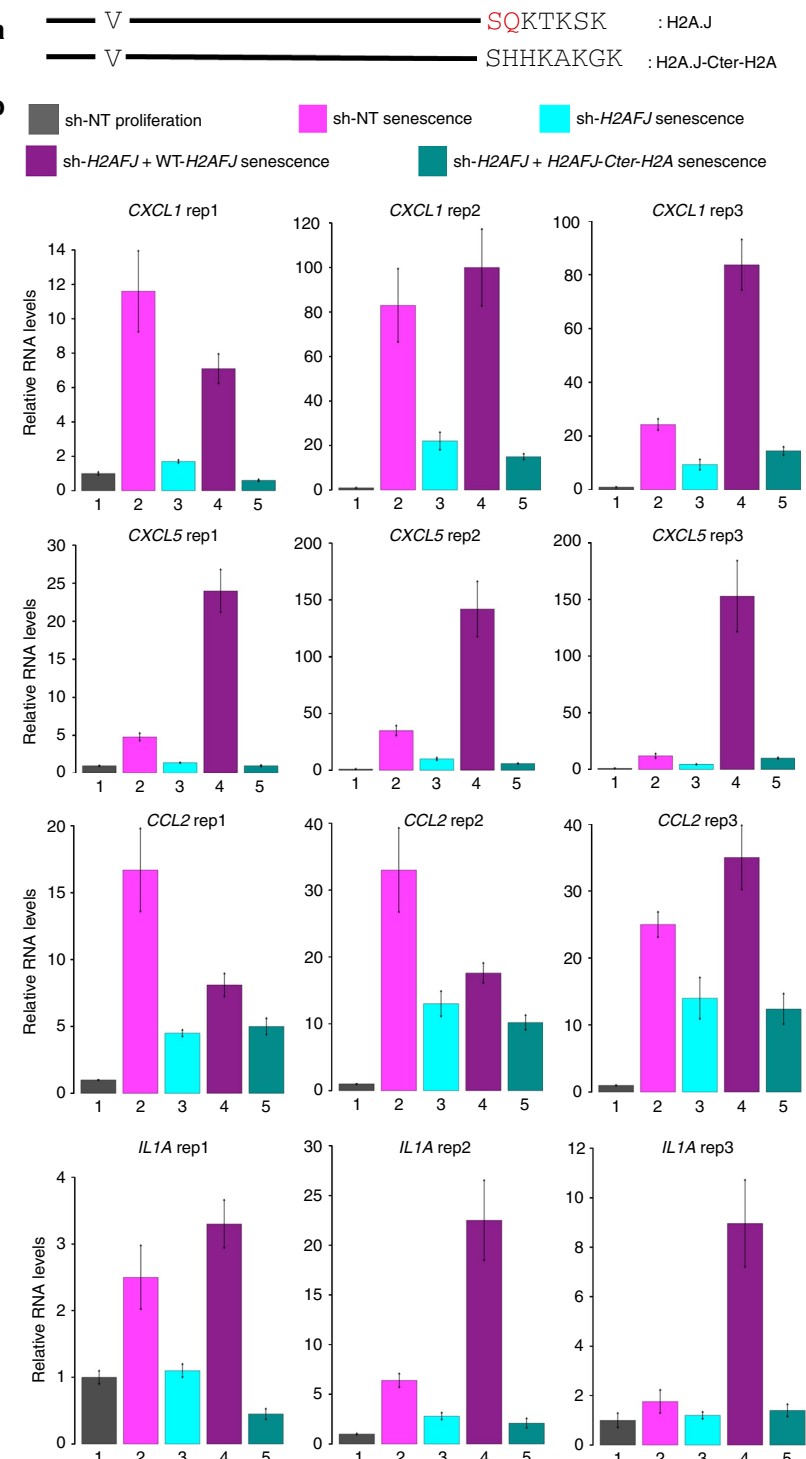

**Figure 6 | The H2A.J C-terminal sequence is required for its function.** WI-38 hTERT cells were infected with pGIPz lentiviruses expressing a No Target shRNA (sh-NT) or an sh3-*H2AFJ* RNA. The sh3-*H2AFJ* cell line was super-infected with pTRIPz lentiviruses expressing sh3-resistant *H2AFJ* cDNAs encoding WT-H2A.J or a mutant H2A.J in which the C-terminus of H2A.J was substituted with the C terminus of canonical H2A type 1 (H2A.J-Cter-H2A). Both the WT and mutant H2A.J contained an N-terminal Flag-HA tag. RNA was isolated from cells induced into senescence by etoposide treatment for 3 weeks and compared to levels in proliferating sh-No Target cells. (**a**) Schematic of V11 and C-terminal aa sequences of H2A.J and H2A.J-C-ter-H2A. (**b**) RT–qPCR quantification of inflammatory gene expression for three biological replicates (independent cultures at different dates) for the indicated genes. The induction ratio for these genes in senescent versus proliferating sh-NT cells varied between biological replicates. The extent to which ectopic WT-H2A.J restored defective gene expression in the sh-*H2AFJ* knockdown cells also varied between biological replicates. However, the H2A.J-Cter-H2A mutant was ineffective at restoring inflammatory gene expression, indicating the functional importance of the specific H2A.J C-terminal sequence. For each biological replicate, the mean and s.d. are shown for three qPCR technical replicates.

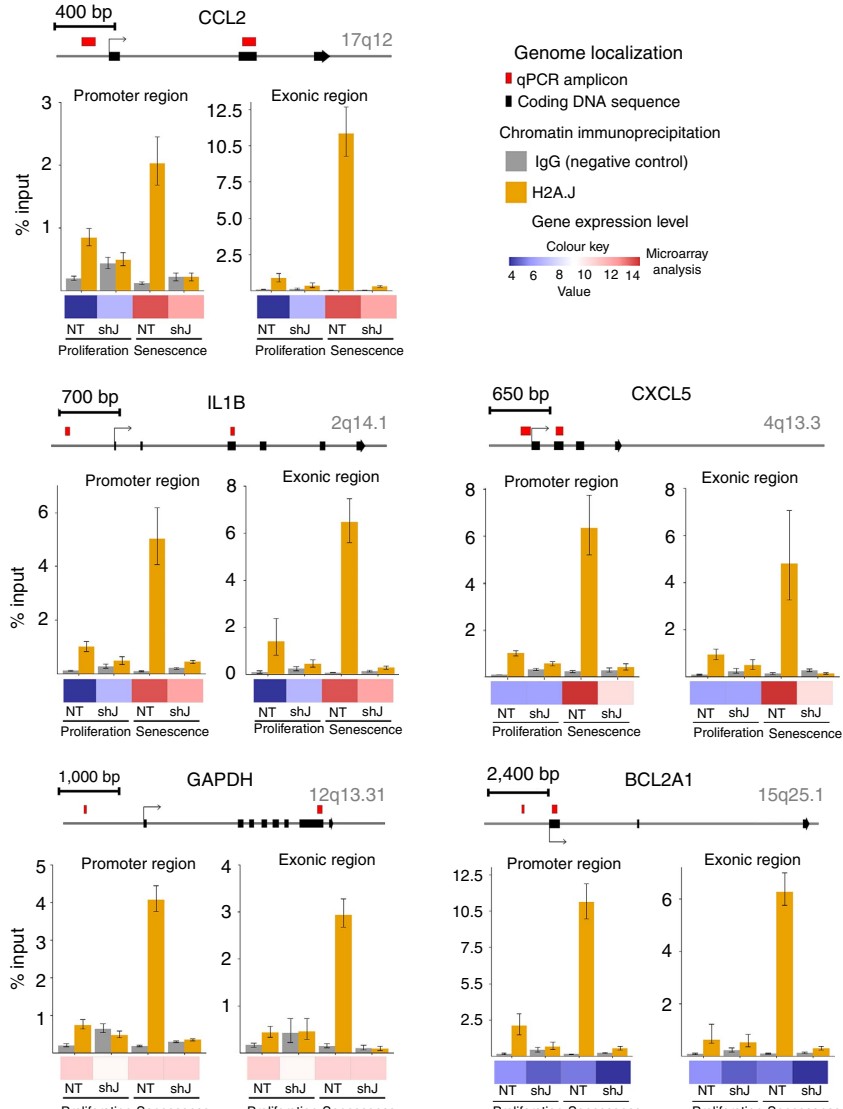

**Figure 7 | ChIP of H2A.J on selected SASP and non-SASP genes.** *CCL2, IL1B*, and *CXCL5* (SASP) and *GAPDH, BCL2A1* (non-SASP) genes were selected for targeted ChIP from cells in proliferation or etoposide-induced senescence. H2A.J was immunoprecipitated from sonicated crosslinked chromatin from WI38hTERT fibroblasts expressing an sh-NoTarget RNA (NT) or an sh3-*H2AFJ* RNA (shJ). The red rectangles indicate the position of the primers used for determining the percentage of DNA immunoprecipitated relative to the input. The sites chosen in the promoter regions of the SASP genes contain NF-kB binding motifs. The scale bars below the figures show gene expression heat maps indicating that H2A.J contributes to the increased expression of these SASP genes in senescence. *GAPDH* shows relatively uniformly high expression in proliferating and senescent cells whereas *BCL2A1* shows relatively uniformly low expression. The percent input is shown as the mean ± s.d. of 3 biological replicates.

verified that H2A.J is found at low levels in proliferating mouse embryonic fibroblasts (MEFs) and accumulates in MEFs in replicative or etoposide-induced senescence (Fig. 8a, Supplementary Fig. 11). We next examined levels of H2A.J in the brain, liver, and kidney of mice at 3 days, 3 months, and 1 year of age. H2A.J was only present at ≈0.7% of all canonical H2A species in all three organs 3 days after birth, but accumulated with age to different extents in an organ-specific manner (Fig. 8b–d). H2A.J levels remained low in the brain through one year (≈6% of canonical H2A), whereas they reached ≈11% of all canonical H2A in the liver at 3 months. H2A.J accumulated even faster in the kidney and reached ≈16% of canonical H2A at 3 months. H2A.J levels appeared to increase slightly in organs from 3 months to 1 year. These results were confirmed by western blotting (Fig. 8e). The canonical H2A type 3 also increased in abundance in these three organs over 1 year of aging at the expense of canonical H2A type 1 that was most

abundant at 3 days after birth. *H2AFJ* mRNA levels were well correlated with H2A.J protein levels in the different organs in 1-year-old mice, suggesting that tissue-specific *H2AFJ* gene expression is a determinant of organ-specific differences in H2A.J levels (Fig. 8f). The early accumulation of H2A.J in some organs suggests that it has tissue-specific functions independent of cellular senescence. It nevertheless remained possible that H2A.J might be a biomarker of senescence in cell types that do not accumulate H2A.J at early ages. We developed an antibody to the unique C terminus of H2A.J, and we showed that it readily distinguishes by immunofluorescence senescent fibroblasts from proliferating or quiescent fibroblasts in mixing experiments (Supplementary Fig. 12). The specificity of this antibody was validated by multiple criteria (see Methods section). H2A.J was seen as weak punctate staining in the nucleus of proliferating cells, and the signal increased strikingly in senescent cell nuclei (Supplementary Fig. 12). We then examined by

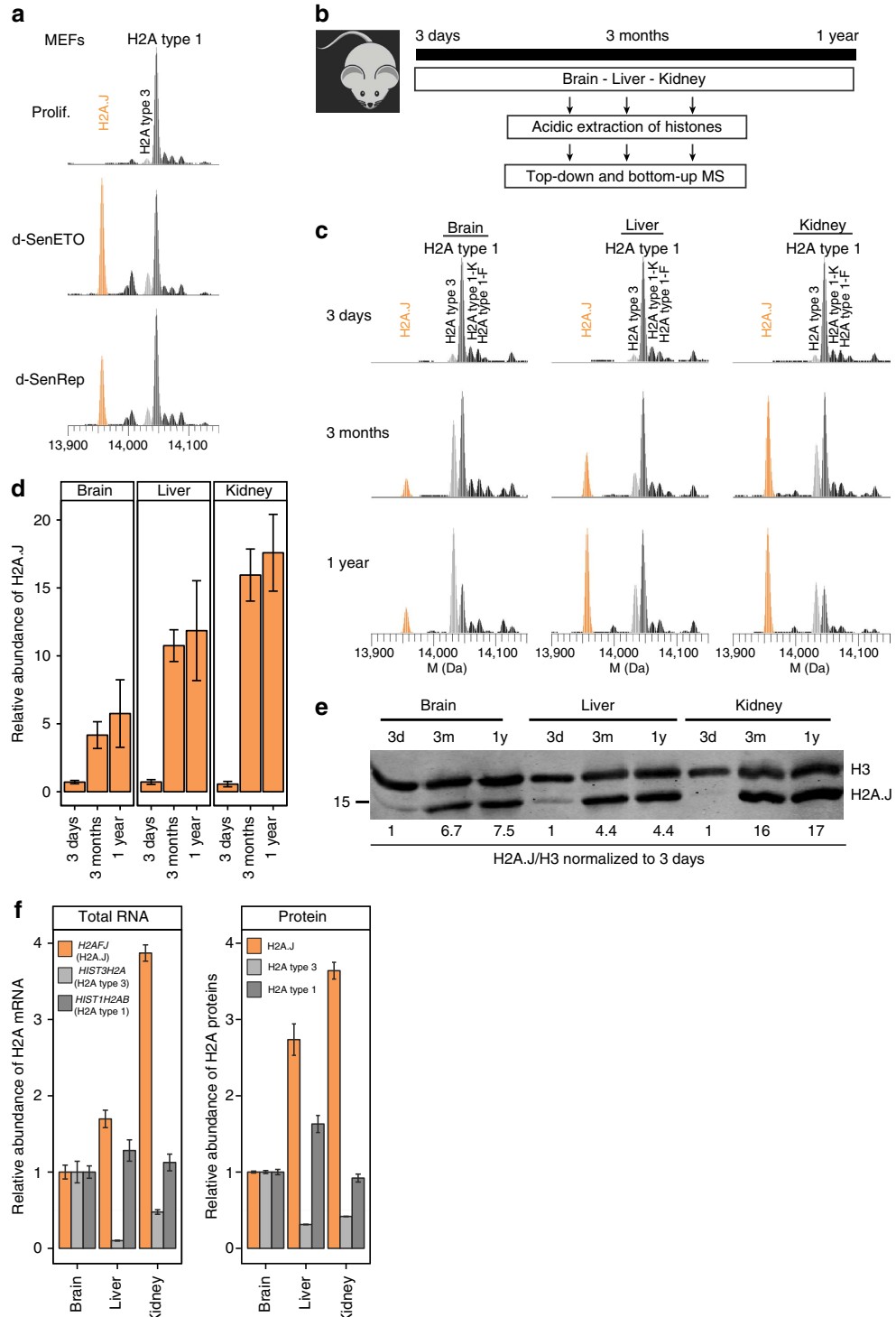

**Figure 8 | Mouse H2A.J accumulates with age in an organ-specific fashion. (a)** H2A.J is present at low levels in proliferating mouse embryonic fibroblasts (MEFs) and increases greatly in replicatively senescent MEFs (d-SenRep) or MEFs maintained in etoposide-induced senescence for 20 days (d-SenETO). **(b)** Experimental timeline for the extraction of histones from the indicated C57BL/6 mouse organs. **(c)** Top-down deconvoluted mass spectra of intact histone H2A2 proteins extracted from mouse organs at the indicated ages. H2A.J is shown in orange and canonical H2A types 1 and 3 are shown in shades of grey. See Supplementary Table 2 for the observed and theoretical masses of these peaks and Supplementary Fig. 11 for their sequence features. **(d)** Relative abundance of H2A.J as the percent of all canonical H2A species by quantifying the Gly5-Arg12 tryptic peptide from H2A.J versus the same peptide from all other canonical H2A species (mean and s.d. for two biological replicates). **(e)** H2A.J and H3 Western blot of acid-extracted histones from the indicated mouse organs. **(f)** qRT–PCR analysis of *H2AFJ* (coding for H2A.J), *HIST3H2A* (coding for H2A type 3), and *HIST1H2AB* (coding for H2A type 1) mRNA levels in 1 year old mouse organs (three technical replicates). Random primers were used to prepare cDNA to quantify total RNA levels (left panel). The right panel shows the quantification of H2A.J, H2A type 3, and H2A type 1 proteins by top-down mass spectrometry from two biological replicates (mean and s.d.). The levels of H2A mRNA and proteins in 1-year-old mouse organs were normalized to the levels found in the brain.

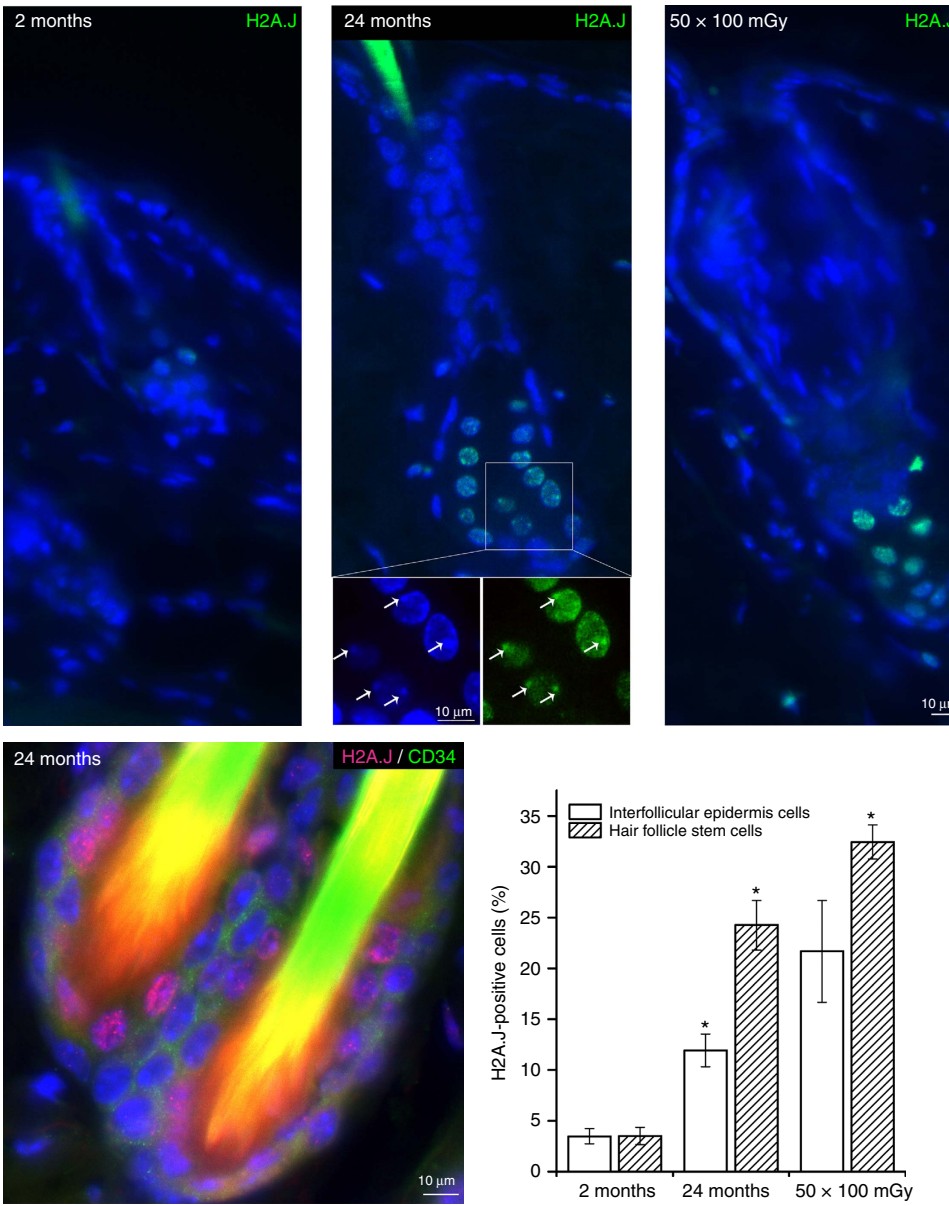

**Figure 9 | H2A.J in mouse hair follicle stem cells and epidermis increases with age and irradiation.** H2A.J accumulates in C57BL/6 mouse hair follicle stem and interfollicular epidermal cells at late ages (24 months) and after repeated low-dose ionizing irradiation (50 × 100 mGy) of young mice (2 months) as seen by immunofluorescence with anti-H2A.J antibodies (green) overlaid on the nuclear DNA stained with Hoechst (blue). The inserts in the 24-month sample show individual staining of nuclear DNA or H2A.J to demonstrate that H2A.J foci correspond to heterochromatic DNA foci. An example of colocalization with the CD34 surface marker displayed by hair follicle stem cells is shown in the lower panel. Counting was performed until 40 H2A.J-positive cells or 200 cells in total were registered for each skin sample. Analysis was performed with three biological replicates, and the one-sided Mann–Whitney $U$-test was used to determine the statistical significance ($P \leq 0.05$). Error bars represent mean ± s.e.

immunofluorescence the expression of H2A.J in hair follicle stem cells (HFSC) and interfollicular epidermal cells (IEC) of young (2 months) and old (24 months) mice, and young mice that were irradiated by fractionated low-dose irradiation (50 × 100 mGy) to induce DNA damage. Less than 5% of cells stained positively for H2A.J in young mice, but this number increased significantly to 25–30% of HFSC and 10–20% of IEC in old and irradiated mice (Fig. 9). The specificity of the immunofluorescent signal was verified by its competition with an excess of antigenic peptide to the C-terminus of H2A.J, but not with a peptide from the C-terminus of canonical H2A (Supplementary Fig. 13).

We next examined sections of human skin at different ages for immunofluorescent staining with H2A.J, Ki-67 and 53BP1. Colocalization studies showed mutually exclusive staining of

epidermal cells with the proliferation marker Ki-67 and H2A.J (Fig. 10a). Ki-67 positive epidermal cells decreased with aging whereas H2A.J positive cells increased. 53BP1 is recruited to chromatin in the form of foci in cells in response to DNA damage or perturbed heterochromatin[34]. The percentage of epidermal cells that contained 53BP1 foci increased from 15% to nearly 30% in skin from 17- to 61-year-old individuals (Fig. 10b). Strikingly, the percentage of H2A.J positive epidermal cells increased from 15% (17-year-old skin) to 60% (61-year-old skin). Although both markers increase in the aging epidermis, the H2A.J pan-nuclear staining is much easier to detect than the small number of discrete 53BP1 foci. H2A.J accumulation may thus represent a novel biomarker of some senescent stem and skin cells in mice and humans.

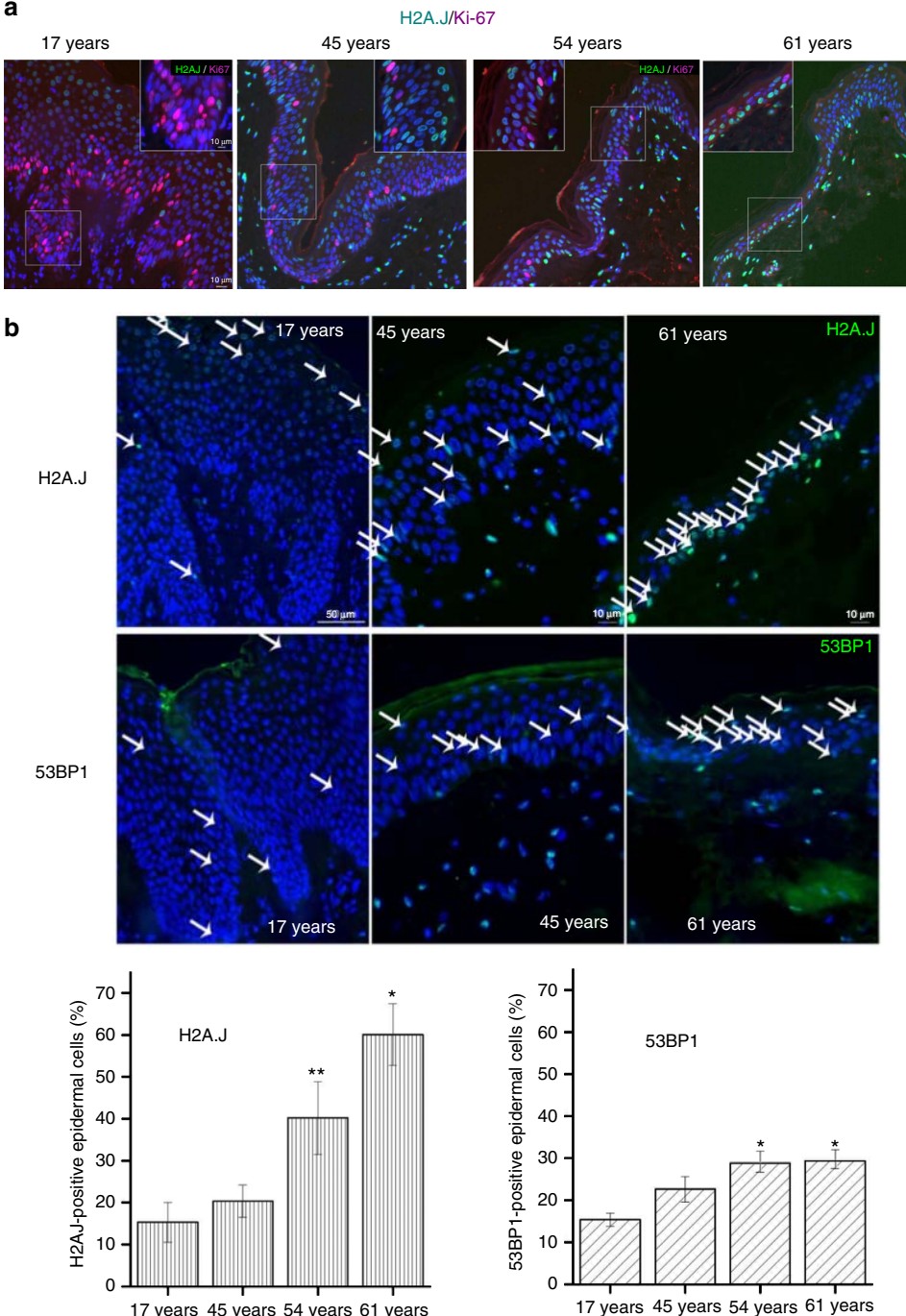

**Figure 10 | H2A.J accumulates in the aging human epidermis.** (**a**) H2A.J and Ki-67 co-localization in sections of human skin of the indicated ages by immunofluorescent staining. H2A.J showed mutually exclusive staining with the Ki-67 proliferation marker. (**b**) H2A.J and 53BP1 foci increase in aging human epidermis. Arrows indicate positively-stained nuclei. A quantification of positive cells for three biological replicates is shown below the immunofluorescence images, and the one-sided Mann–Whitney $U$-test was used to determine the statistical significance (*$P < 0.05$, **$P < 0.01$). Error bars represent mean ± s.e.

## Discussion

The H2A protein family has 19 members[17]. Eleven of these are canonical type 1,2,3 histones with a small number of amino acid differences. Most canonical histones are coded for by clustered genes on human chromosomes 1 and 6 that are expressed preferentially in S-phase for packaging newly-replicated DNA. Eight H2A family members (H2A.V, H2A.X, H2A.Z, H2A-Bbd type 1, H2A-Bbd type 2/3, macro-H2A.1 and macro-H2A.2) are clearly diverged variants with specialized functions, and are expressed independently of DNA replication[11,35]. The unique

*H2AFJ* gene encoding H2A.J was previously shown to be expressed independently of DNA replication[18,24], but this study is the first, to the best of our knowledge, to describe H2A.J at the protein level. H2A.J differs from canonical H2A by only five amino acids involving a substitution of Val for Ala at position 11 in the N-terminal tail, and a unique C terminus containing a potentially phosphorylatable SQ motif (Fig. 1d). Its recent evolution (Supplementary Fig. 3) suggests that it may have mammalian-specific functions. Using mass spectrometry and a specific antibody, we found that H2A.J is expressed at low levels

in the chromatin of proliferating or quiescent (serum-starved) fibroblasts (around 1% of canonical H2A), but accumulates in senescent fibroblasts containing persistent DNA damage (around 10% of canonical H2A) (Fig. 1c). The correlation with persistent DNA damage (Fig. 1e) suggests that DNA-damage signalling may regulate H2A.J levels. *H2AFJ* mRNA was not expressed at higher levels in senescent cells compared to quiescent cells (Fig. 1g), but H2A.J protein levels are much higher in senescent cells with DNA damage compared to quiescent cells (Fig. 1b,c,f), so H2A.J accumulation in senescence must involve a post-transcriptional mechanism. We also observed differences in the rates of H2A.J protein accumulation with age in the mouse brain, liver, and kidney that correlated with differences in *H2AFJ* RNA levels (Fig. 8). Thus, H2A.J expression can be regulated at both the transcriptional and post-transcriptional levels. Systematic immuno-staining of tissue sections will be required to identify the cell types that express H2A.J and to investigate the basis of its tissue-specific differences in gene expression.

H2A.J depletion did not affect the induction or the maintenance of senescence induced by etoposide treatment (Supplementary Fig. 6), but it did strongly inhibit the expression of immune response and inflammatory genes, including many associated with the SASP (Figs 4 and 5). The SASP is implicated in signalling the presence of senescent cells to the immune system and in wound healing[1,6]. These genes are highly repressed in proliferating cells to prevent inappropriate inflammatory signals in the absence of stress, but they are highly expressed in senescent cells. Time-course experiments revealed that full derepression of these genes is a slow process occurring over 2 weeks during senescence (Fig. 5d). These findings are in agreement with previous work showing that expression of the SASP is a slow process that takes more than a week for peak expression in fibroblasts induced into senescence by DNA damage[36]. Regulation of SASP gene expression is complex with multiple parallel amplification pathways and inhibitory feedback loops[13,37]. In senescence induced by DNA damage, both persistent DNA damage signalling and p38 MAP kinase activation were required for SASP gene expression mediated by the NF-kB transcription factor[30,36]. Activation of ATM and Chk2 occurred rapidly on DNA damage, but was insufficient for SASP expression. Activation of the p38 MAP kinase was also required for SASP expression, but this activation occurred several days subsequent to senescent-inducing DNA damage by an unknown mechanism[38]. The delayed activation of p38 contributes to the delayed expression of the *SASP* genes. We found that knock-down of H2A.J also delayed and inhibited SASP gene expression (Fig. 5). Targeted chromatin immunoprecipitation experiments and genome-wide ChIP-seq revealed that H2A.J occupancy increased in senescence at inflammatory genes whose expression is dependent on H2A.J, as well as on genes whose expression is unaffected by H2A.J depletion (Fig. 7, Supplementary Fig. 10). The specificity of H2A.J action thus cannot be due to its preferential deposition at inflammatory genes. This result suggests that other factors must act in concert with H2A.J to promote strong inflammatory gene expression. The distinct C termini of H2A variants contribute to their functional specificities[35]. The unique H2A.J N and C-termini may affect chromatin compaction and interact differentially with protein partners. Our complementation experiments indicated that the canonical H2A C terminus could not functionally replace the H2A.J C terminus, supporting an essential role for the H2A.J C-terminal sequence (Fig. 6). It is also possible that H2A.J accumulation in senescence may act by displacing other repressive H2A variants. Further mechanistic work will be required to understand how persistent DNA damage facilitates H2A.J accumulation and how H2A.J stimulates SASP gene

expression. Study of this pathway may lead to novel approaches to decrease chronic inflammation associated with the accumulation of senescent cells during aging.

*H2AFJ* overexpression in proliferating cells increased the expression of many inflammatory and immune-response genes (Fig. 4e). The *H2AFJ* gene was amplified and over-expressed in some breast cancers[28,29] and other cancers (see the COSMIC database: http://cancer.sanger.ac.uk/cosmic/gene/analysis?ln=H2AFJ). *H2AFJ* over expression in these cancers may have been selected for during tumour progression because of its ability to promote inflammatory gene expression that is pro-tumorigenic[27].

The accumulation of H2A.J in senescent fibroblasts with persistent DNA damage suggested that it might be a potential biomarker for this state. Our immune staining of mouse skin showed that H2A.J was present at low levels in hair follicle stem cells and interfollicular epithelial cells of young mice (2 months), but increased significantly in these cells in old mice (2 years) or after repeated low-dose irradiation of young mice (Fig. 9). H2A.J also increased dramatically in epidermal cells of aging human skin. The strong pan-nuclear staining of H2A.J was much easier to score compared to 53BP1 foci in the aging epidermis (Fig. 10). H2A.J thus appears to be a novel biomarker for some senescent stem and skin cells. Mass spectrometry analyses also indicated that H2A.J remained at low levels in the mouse brain through one year of age. However, although present at low levels (0.7% of canonical H2A) at birth, we also observed rapid accumulation of H2A.J in the liver and kidney to reach about 11% and 16% of canonical H2A by 3 months of age (Fig. 8c,d). The early accumulation of H2A.J in the liver and kidney indicates that it very likely has tissue-specific functions that are independent of senescence. Such functions would be best studied in the future with an *H2AFJ* knock-out mouse model.

## Methods

**Cell culture.** WI-38 human embryonic fibroblasts from the ATCC were immortalized with hTERT and grown in MEM (Invitrogen #21090) + 10% Fetal Bovine Serum + 1 mM Sodium Pyruvate + 2 mM L-glutamine + 0.1 mM non-essential amino acids[16,21]. Cells were mycoplasma negative. Senescence was induced by incubating cells with 20 µM etoposide, or passing non-immortalized cells until replicative senescence, or by inducing a constitutively active form of the RAF1 kinase fused to GFP and the estrogen-receptor binding domain (GFP-RAF1-ER) by the addition of 20 nM 4-hydroxy-tamoxifen (4-HT) to cells[16,21]. IMR90-hTERT/ER-Rasval12 were also induced into senescence by addition of 100 nM 4-HT to the medium[16]. SA-β-gal assays were performed by staining formaldehyde-fixed cells with 1 mg ml$^{-1}$ 5-bromo-4-chloro-3-indolyl-β-D-galactoside (X-Gal) in 40 mM citric acid-sodium phosphate (pH 6.0), 5 mM potassium ferrocyanide/5 mM potassium ferricyanide, 150 mM NaCl, 2 mM $MgCl_2$ buffer overnight at 37°C and γH2AX immunofluorescence assays were performed with mouse anti-γH2AX antibodies (Millipore 05-636) diluted 1/500 with an incubation of 1 hour at room temperature as described[16,21]. Quiescence was induced by serum limitation by incubating cells with medium containing 0.1% Fetal Bovine Serum. Mouse embryonic fibroblasts were grown to replicative senescence in 5% $CO_2$ at ambient oxygen. Senescence was also prematurely induced by incubating proliferating MEFs with 2.5 µM etoposide for 20 days.

**shRNA knock-down of H2AFJ.** pTRIPz-sh-No Target (Thermo RHS4743), pTRIPz-sh2-*H2AFJ* (Thermo V3THS_302395) pTRIPz-sh3-*H2AFJ* (Thermo V3THS_351902), and pGIPz-sh-NoTarget (RHS4346) were purchased from Thermo Scientific. The sh2-*H2AFJ* RNA targets positions 210-228 of the human *H2AFJ* coding sequence: 5'-CGCGCGTGACAACAAGAAG-3'. The sh3-*H2AFJ* RNA targets positions 265–283 of the human *H2AFJ* coding sequence: 5'-CGCAACGACGAGGAGTTAA-3'. The sh3-*H2AFJ* sequence was also transferred to pGIPz. Another pTRIPz-sh-*H2AFJ* RNA (Thermo V2THS_156605), that we called sh1, targets a putative 3'-UTR sequence that is not transcribed in normal human fibroblasts and was not effective at knocking-down *H2AFJ*.

An sh2-resistant coding sequence with an N-terminal Flag-HA epitope: 5'-ATGGACTACAAGGACGACGATGACAAGCTCGATGGAGGATACCCCTA CGA CGTGCCCGACTACGCCGGAGGAATGTCCGGTCGCGGGAAACAGGG CGGCAAAGTGCGAGCAAAGGCCAAATCCCGCTCCTCCCGCGCGGGCCTG CAGTTCCCGGTGGGCCGAGTGCACAGACTGCTGCGGCAAAGGGAACTACG CGGAGCGAGTGGGCGCCGGGGCGCCGGTGTACCTGGCGGCGGTGTTGG

AGTACCTTACGGCGGAGATCCTGGAGCTGGCTGGCAACGCTGCCAGGGA
TAATAAAAAAACCAGGATAATTCCCCGCCACCTGCAGCTCGCCATCCGC
AACGACGAGGAGTTAAACAAGCTGCTGGGCAAAGTGACCATCGCTCAGG
GCGGCGTCCTGCCCAACATCCAGGCCGTGCTGCTGCCCAAGAAGACGGA
GAGTCAGAAGACGAAGAGCAAATGA-3′ was synthesized by Eurofins Genomics and cloned as an *Age*I–*Mlu*I fragment in the pTRIPz vector. The sh2-resistant coding sequence contains multiple silent mutations in the sequence targeted by sh2. Lentiviruses were prepared from pTRIPz plasmids by co-transfection of 293T cells with packaging plasmids as per standard techniques[39]. WI38hTERT cells were infected with lentiviruses and stable cell populations were selected for puromycin resistance conferred by pTRIPz.

**MS profiling of core histones.** The profiling of core histone variants was performed by ultra-high performance reverse-phase liquid chromatography coupled to mass spectrometry (UHPLC-MS) of acid-extracted histones either as is for histones from fibroblasts in culture, or after mild performic oxidation for histones extracted from mouse organs as previously described[15,16]. Briefly, oxidized histones were loaded and separated on a C18 Hypersil GOLD column (2.1 × 150 mm, 175 Å, 1.9 μm, Thermo) at a flow rate of 300 μl min$^{-1}$ with a linear gradient of 0–80% B in 13.5 min (with solvent A: $H_2O$ containing 0.1% formic acid and solvent B: ACN containing 0.1% formic acid). MS acquisition was performed on an LTQ-Orbitrap Discovery mass spectrometer (Thermo, San Jose, CA) operating in the positive ion mode (acquisition from $m/z$ 500 to 2,000) using a resolution set at 30,000 (at $m/z$ 400). The resulting mass spectra were deconvoluted using the Xtract software included in the Xcalibur package (Thermo). Proteins were identified by their mass as indicated in Supplementary Tables 1 and 2. Relative quantification of modified histone forms and variants was performed on deconvoluted mass spectra by dividing the intensity of a given MS peak by the sum of the intensities of the different MS peaks composing the considered spectrum. Note that the profiles shown in Figs 1b and 8c correspond to fractions of the C18 column containing H2A2 (H2A fraction 2) histones with leucine at position 52 (numeration starting from Met1). The H2A1 (H2A fraction 1) histones with methionine at position 52 elute at a distinct position (Supplementary Figs 1 and 2). Thus, to quantify the amount of H2A.J versus all other canonical H2A histones, we digested total histones with trypsin as described below, and we compared the area for the Gly5-Arg12 tryptic peptide peak from H2A.J versus the same peptide from all other canonical H2A species. H2A.J contains Val11 whereas all canonical H2A species contain Ala11 (numbering relative to Met1; Fig. 1d and Supplementary Figs 2 and 10). The relative abundance of H2A.J versus canonical H2A was thus calculated as: (Area H2A.J peptide/(Area H2A.J peptide + Area canonical H2A peptide)) × 100.

**MS analysis of histone tryptic digests.** Acid-extracted histones were first *in vitro* propionylated on lysine residues with propionic anhydride, then digested with trypsin, and finally subjected to a second round of propionylation to block the newly formed N-terminal residues[15,40]. Analyses of the resulting tryptic digests were then performed on an LTQ-Orbitrap Discovery mass spectrometer that was operated in the data-dependent acquisition mode, using the five most intense precursor ions. The MS survey scan was performed from $m/z$ 300 to 2,000 in the Orbitrap at a resolution of 30,000 (at $m/z$ 400), while the fragment ions were detected in the linear ion trap after collision-induced dissociation (CID).

**Anti-H2A.J rabbit polyclonal antibodies.** The C-terminal H2A.J peptide CTESQKTKSK was covalently linked at its N-terminal cysteine to the carrier protein bovine serum albumin through a maleimide linkage and injected into two rabbits. The N-terminal cysteine is not found in H2A.J but was added to facilitate cross-linking of the peptide. After 3 boosts, serum was collected and anti-H2A.J antibodies were purified by binding to the C-terminal H2A.J peptide attached to the Sulfolink resin (Pierce) and eluting with 100 mM glycine (pH 2.5) followed by immediate neutralization with a 1/20 volume of 1M Tris (pH 8.5).

Please note that several commercial antibody providers sell an 'anti-H2A.J' antibody using as immunogen a peptide shared with canonical H2A or the full-length H2A.J. It is highly unlikely or impossible that these antibodies specifically recognize H2A.J.

**Validation of the specificity of the purified anti-H2A.J Ab.** Multiple independent criteria were used to verify the specificity of the anti-H2A.J antibody as summarized here:

Figures 1, 2 and 8: the intensity of the H2A.J band in western blots is in excellent agreement with the amount of H2A.J determined by mass spectrometry in multiple conditions, in several cell lines, and in several mouse organs.

Supplementary Figs 5A,B: two independent shRNAs were identified that are effective in knocking-down *H2AFJ* mRNA. The H2A.J band identified by our antibody in Western blots is greatly decreased in the cells expressing these shRNAs. Furthermore, when we ectopically express a Flag-HA-tagged cDNA encoding H2A.J, we observe a novel band that migrates at the expected size for the Flag-HA-H2A.J.

Supplementary Fig. 7: cells expressing an shRNA to *H2AFJ* showed reduced staining with our anti-H2A.J Ab when compared to sh-NoTarget cells.

Supplementary Fig. 9B: the H2A.J antibody directed to the H2A.J C-terminus does not recognize an H2A.J mutant in which the C terminus of H2A.J was exchanged for the C-terminus of canonical H2A type 1.

Supplementary Fig. 12: senescent cells show much higher immunofluorescent staining with our anti-H2A.J antibody compared to proliferating cells, in agreement with the abundance difference observed by mass spectrometry.

Supplementary Fig. 13: the anti-H2A.J immunofluorescence signal observed in formalin-fixed mouse tissue sections is completely eliminated by competition with a molar excess of the H2A.J C-terminal peptide used for immunization, but not with a C-terminal canonical H2A peptide.

Supplementary Fig. 14: an Elisa peptide competition assay shows that the anti-H2A.J Ab has 1,000-fold higher affinity for the H2A.J C-terminal peptide compared to a canonical H2A C-terminal peptide.

**Radiation schedule and tissue sampling in mice.** Young (2-month-old) and aged (24-month-old) C57BL/6 mice were purchased from Charles River Laboratories (Sulzfeld, Germany). Mice were housed four to six per cage in pathogen-free rooms (temperature 22 ± 2 °C, humidity 55 ± 10%, 12 h light–dark cycle) to minimize the risk of infections and supplied with standard laboratory diet and water ad libitum. Serological assessment was conducted at least quarterly to test for infection, and all tests were negative.

To evaluate the effects of repeated low-dose radiation with 50 × 100 mGy, C57BL/6 mice were irradiated for 10 weeks once daily from Monday to Friday (with 24 h between exposures). For the whole-body irradiation in a linear accelerator (Artiste, Siemens, Munich, Germany), the animals were placed in an 18 cm diameter Plexiglas cylinder covered by 1.5 cm thick plastic to improve photon dose homogeneity. The radiation characteristics were as follows: size of the radiation field, 30 × 30 cm; collimator angle 0°; gantry angle 0°; source surface distance, 208 cm; beam energy, 6 MV photons; dose-rate, 2 Gy min$^{-1}$. Computed-tomography-based 3-dimensional dose calculations were made with the Pinnacle planning system (Philips Radiation Oncology Systems, Fitchburg, WI, USA). 72 h after the last radiation exposure, mice were anesthetized and the dorsal skin removed, fixed overnight in 4% paraformaldehyde (Sigma-Aldrich Chemie GmbH, Munich, Germany) and processed for further analysis. The experimental protocol was approved by the Medical Sciences Animal Care and Use Committee of the University of Saarland.

**Tissue sampling in humans.** Human skin samples were collected from voluntary donors of varying ages. Skin biopsies were obtained from the abdomen from photo-protected areas of the skin during routine surgical procedures. Skin samples were fixed overnight in 4% paraformaldehyde (Sigma-Aldrich Chemie GmbH, Munich, Germany) and processed for further analysis. None of the donors had any chronic diseases or used any medications. Protocol procedures were approved by the local ethics committee ('Ethikkommission der Ärztekammer des Saarlandes'), and all donors provided written informed consent.

**Immunofluorescence analysis of tissue sections.** Formalin-fixed tissues were embedded in paraffin and sectioned at a thickness of 4 μm (ref. 34). After deparaffinization by three 7-min rounds in xylene and rehydration in decreasing alcohol concentrations (100, 96, 90, 80, 70% for 3 min each), sections were boiled in citrate buffer, pH 6 (Dako Agilent pathology solutions, Santa Clara, CA , USA) for 1 h at 96 °C and incubated with Roti-Immunoblock (Carl Roth, Karlsruhe, Germany). Sections were incubated with primary antibodies (anti-H2A.J, 1:200; anti-Ki67 (eBioscience 14-5698-82), 1:200, Thermo Scientific, Darmstadt, Germany; anti-53BP1 (NB100-304), 1:5,000, Novus biologicals, Littleton, CO, USA) overnight at 4 °C, and anti-CD34 (553731), 1:50, BD Biosciences, Heidelberg, Germany, 1h at RT), followed by AlexaFluor-488 or AlexaFluor-568 secondary antibodies (1:400, Invitrogen, Karlsruhe, Germany) for 1h at RT. Finally, sections were mounted in VECTAshield with 4′,6-diamidino-2-phenylindole (DAPI; Vector Laboratories, Burlingame, USA). For quantitative analysis, H2A.J-positive cells were counted visually under a Nikon E600 epifluorescent microscope (Nikon, Düsseldorf, Germany). Counting was performed until at least 40 H2A.J positive cells or 200 cells in total were registered for each skin sample. Accordingly, in aged or low-dose irradiated mice we analyzed 200–400 cells in each skin section, and averaged the results obtained from three different mice for each data point. We calculated the percentage of 53BP1-positive cells by counting at least 40 53BP1 foci with a minimum 100 cells. The one-sided Mann–Whitney $U$-test was used to determine the statistical significance ($P \leq 0.05$).

**Western blots.** Acid-extracted histones or whole-cell extracts were electro-phoresed in SDS-15% polyacrylamide gels and proteins were transferred to nitrocellulose membranes. Membranes were blocked and incubated with antibodies using the Li-Cor Odyssey infrared imaging system following the manufacturer's protocols. Rabbit anti-H2A.J antibody, anti-H3 (Abcam ab1791), anti-GAPDH (Abcam ab9484), and anti-HA.11 (mouse monoclonal 16B12) antibodies were all used at 1/1,000 dilutions for 1 hour incubations with membranes at room temperature. Uncropped western blots are shown in Supplementary Fig. 15.

**Illumina Bead Chip Array transcriptome analyses.** WI38hTERT cells stably infected with pTRIPZ-sh-No Target, pTRIPz-sh2-*H2AFJ*, pTRIPz-sh3-*H2AFJ*, or pTRIPz-sh2-*H2AFJ* + pTRIPz-sh2-resistant-*Flag-HA-H2AFJ* were grown in the presence of 1 μg ml$^{-1}$ doxycycline for 4 days to express the shRNAs and to allow time for knock down of *H2AFJ* mRNA. One set of cells was collected for RNA extraction under proliferation conditions. Another set of cells was then treated with 20 μM etoposide for 2 weeks with change of medium after one week to induce stable senescence. After 2 weeks in etoposide, fresh medium without etoposide was added and cells were incubated for one additional week before collecting cells under deep senescence conditions. Cells remained in senescence with persistent DNA damage under these conditions (Supplementary Figs 6 and 7). Doxycycline was maintained throughout the 3 weeks in senescence to ensure continual expression of the shRNAs. Five biological replicates were analyzed for each cell type under proliferative conditions except for the pTRIPZ-sh3-*H2AFJ* cells that were represented by four biological replicates. For etoposide-induced senescence, eight biological replicates of pTRIPZ-sh-No Target cells, four biological replicates of pTRIPz-sh2-*H2AFJ* cells, and five biological replicates of pTRIPZ-sh3-*H2AFJ* and pTRIPZ-sh2-*H2AFJ* + pTRIPZ-sh2-resistant-*Flag-HA-H2AFJ* cells were prepared and analyzed.

RNA was extracted from 500,000 cells using Nucleospin RNA kits (Machery-Nagel). The total RNA yield (ng) was determined spectrophotometrically using the NanoDrop ND-100 (Labtech, France). Total RNA profiles were recorded using a Bioanalyzer 2100 (Agilent). RNA integrity numbers were determined, and the mean value found was $9.8 \pm 0.24$ s.d. The coefficient of variation (CV) was 2.46%. cRNA was synthesized, amplified, and purified using the Illumina TotalPrep RNA Amplification Kit (Life Technologies, France) following the manufacturer's instructions. Briefly, 200 ng of RNA were reverse transcribed. After second strand synthesis, the cDNA was transcribed *in vitro* and cRNA labelled with biotin-16-UTP. Labelled probes were then hybridized with beads using Illumina Bead Chip human HT-12 v4 arrays. These Bead Chips contain 47,231 unique 50-mer oligonucleotides in total, with hybridization to each probe assessed at 15 different beads on average. 28,403 probes (60.7%) are targeted at Reference Sequence (RefSeq) transcripts and the remaining 18,543 (39.3%) are for other transcripts, generally less well characterized (including predicted transcripts).

Bead Chips were scanned on the Illumina iScan Reader using Illumina iScan control software version 3 and Illumina Genome Studio software (version 2011.1) were used for preliminary data analysis. Multiple quality controls were performed and analyzed with the Illumina Genome Studio software, including scatter plots of control RNA samples from different labelling runs. The scatter plots compared duplicated control and experimental samples and were used to calculate correlation coefficients. These comparisons determined whether controls and samples from different labelling runs varied in quality. The control summary report generated by Genome Studio evaluates variations in signal intensity, hybridization signal, background signal, and the background/noise ratio for all the samples analysed in that run. The Illumina idat files have been deposited in the GEO database under accession number GSE62701.

**Bead Chip bio-informatics analysis.** Raw bead chip array intensity data contained in the proprietary Illumina .idat files were analyzed using open-source Bioconductor packages[41]. Data was retrieved from the .idat files using the beadarray package[42]. Data were quantile normalized with control probes using the neqc function[43] in the limma R package. Batch effects were corrected with the sva package[44]. The gene symbols corresponding to Illumina probe ids were retrieved with the Illumina HumanHT12v4 annotation data. Differential gene expression analysis was performed with the limma package[25] and heat maps were constructed with pheatmap or heatmap2. Gene set enrichment analysis (GSEA)[26] was conducted on the sva corrected gene expression matrix (Bead Chip data) using the javaGSEA desktop application provided by the Broad Institute using the default parameters. Nominal *P*-values were calculated based on 1,000 permutations of the gene sets. A 40-member SASP gene set (Fig. 5a,b) was derived from the literature[30,45]: *AREG, CCL2, CCL3, CCL5, CCL8, CCL26, CCR3, CSF2, CXCL1, CXCL2, CXCL3, CXCL5, CXCL6, CXCL8, CCL20, HGF, ICAM1, IGFBP2, IGFBP4, IGFBP5, IGFBP6, IGFBP7, IL1A, IL1B, IL6, IL6ST, IL7, IL13, MIF, MMP1, MMP3, MMP10, MMP12, MMP13, MMP14, OSM, TIMP2, TNFRSF10C, TNFRSF11B* and *TNFRSF18*. Two of these genes (*AREG* and *CXCL3*) were not present as good-quality probes in the Illumina HT-12 v4 Bead Chip arrays and were thus not available for expression analysis.

GO analyses were performed with the goana function within the limma R package[25]. Phylogenetic analyses were done by aligning sequences with ClustalW and constructing trees with the R package APE (Analyses of Phylogenetics and Evolution)[46]. R scripts used for the transcriptomic analyses are available on GitHub (https://github.com/CarlMann/sh-H2AFJ-transcriptome-analysis).

**RNA-seq.** WI38hTERT cells stably transduced with pTRIPZ-sh-No Target or pTRIPz-sh2-*H2AFJ* were grown in the presence of 1 μg ml$^{-1}$ doxycycline for 4 days to express the shRNAs and to allow time for knock-down of *H2AFJ* mRNA. One set of cells was collected for RNA extraction under proliferation conditions. Another set of cells was then treated with 20 μM etoposide for 2 weeks to induce stable senescence. After 2 weeks in etoposide, fresh medium without etoposide was added and cells were incubated for one additional week before collecting cells

under deep senescence conditions. Doxycycline was maintained throughout the 3 weeks in senescence to ensure continual expression of the shRNAs. 3 biological replicates in etoposide-induced senescence and 2 biological replicates in proliferation were analyzed for each cell line.

Ribo-Zero Magnetic Gold Kit (Epicentre, Cat #MRZG126 , Madison, WI) was used to deplete ribosomal RNA. The isolated RNA was then sheared using RNA Fragmentation Reagents (Life Technologies, #AM8740, Grand Island, NY, USA) at 80 °C for 1 min. The RNA fragment size range was mainly 100–500 bp on 2% E-gel (Life Technologies, #G4020-02, Grand Island, NY, USA). Double-stranded cDNA synthesis was primed with random hexamers using a double-strand cDNA Kit (Life Technologies, #11917-010, Grand Island, NY, USA) following the manufacturer's protocol with some modification. Briefly, after first-strand cDNA synthesis, NucAway Spin Column (Life Technologies, Grand Island, NY, USA) was used to remove dNTPs. Then, during the second strand reaction, dUTP was incorporated instead of dTTP. The reaction was then purified with QIAquick PCR spin column (Qiagen, #28106, Valencia, CA). The cDNA was then end-repaired with the End-It kit (Epicentre, Cat# ER0720, Madison, WI). After treatment with Klenow fragment (NEB, #M0212s, Ipswich, MA) and dATP, Illumina adapters were ligated to the protruding 3′-'A' base (Promega Cat #M8221, Madison, WI). Ligated products were size selected on 2% E-gel (250-300 bp). The dUTP-containing second strand was removed by incubating with Uracil-DNA Glycosylase (New England Biolabs, Ipswich, MA, USA), and the libraries were amplified with Phusion high-fidelity PCR master mix (NEB, #M0531 Ipswich, MA) and the Illumina genomic DNA primers 1.1 and 2.1 with the following conditions: 98 °C for 30 s, 15 cycles of (98 °C for 10 s, 65 °C for 30 s, 72 °C for 30 s), 72 °C for 5 min. The final amplified libraries were further purified by agarose gel excision and extraction. The library was subjected to 101 bp paired-end sequencing using Illumina's HiSeq 2000 Sequencer. Approximately 25 million paired-end reads were acquired for each sample. RNA-seq data was deposited at the EBI Array Express under accession number E-MTAB-4920.

**RNA-seq bio-informatic analysis.** Quality control and adapter trimming of the fastq sequence files were done with FastQC (v0.11.2)[47] and the Trim Galore wrapper (v0.4.1)[48] for Cutadapt (v1.8.1)[49]. Options were set to remove Illumina TruSeq adapters and end sequences of quality inferior to 20: trim_galore--illumina--paired--fastqc. Successful processing was verified by re-running FastQC. Sequences were then aligned to the GRCh37 (hg19) human genome sequence using Tophat2 (v2.1.1)[50,51] with default parameters. The accepted_hits.bam files produced by Tophat2 were then used with featureCounts from the Rsubread package[52] to assign and count the mapped fragments to genes using the hg19 annotation table provided in Rsubread. Differential gene expression analysis based on these count tables was then performed with DESeq2 (ref. 53). A script for the R specific portion of this analysis is provided on GitHub (https://github.com/CarlMann/sh-H2AFJ-transcriptome-analysis).

**Targeted chromatin immunoprecipitation.** Cells were were crosslinked with formaldehyde 1% for 10 min and then quenched with glycine at 150 mM final concentration. Cells were washed with PBS, scraped and pelleted by centrifugation at 800 g for 10 min. Cells were resuspended in TM2 buffer (10 mM Tris pH 7.4, 1 mM EGTA (pH 8.0), 2 mM MgCl₂, 0.5 mM PMFS, 1% Roche protease inhibitor cocktail) and transfered into Covaris-specific tubes. Volumes were adjusted to 1 ml. Sonication was then performed with a Covaris ultrasonicator S220 with the following settings: Peak incident Power : 140 W ; Duty Factor : 10% ; Cycles per burst : 200; duration : 20 min. Samples were then centrifuged at 14,000 g for 15 mins. at 4 °C. Chromatin sonication was checked by treating an aliquot with RNase A and proteinase K and then running in a 2% agarose gel to check lengths of the DNA fragments. DNA concentrations were determined using a Nanodrop 2000. The protocol was optimized to yield a peak of sonicated chromatin around 300–500 bp. For each immunoprecipitation, up to 10 μg of chromatin was used. Buffer was added to reach a volume of 500 μl. Chromatin was prewashed using ChIP-grade protein G sepharose beads (GE Healthcare) and Sepharose beads + Normal Rabbit IgG (Cell Signaling #2729). An input of 5% (25 μl) was kept. 6 μg of anti-H2A.J, 1.6 μg of anti-H2A (Abcam 18255), or 5 μg of control rabbit IgG were added and incubated overnight with rotation at 4 °C. ChIP-grade protein G agarose beads were added to each IP reaction and incubated for 2 h. Beads were pelleted with brief centrifugation and supernatants were removed. Beads were washed 8 times in 10 mls low Salt buffer (10 mM Tris pH 7.4, 1% triton X-100, 2 mM MgCl₂, 0.5 mM PMFS, 1% PIC, 120 mM NaCl).

DNA was eluted using TE buffer + 1% SDS and by incubating at 65 °C in a Thermomixer (1,200 r.p.m.) for 30 min. Beads were pelleted with a 1 min centrifugation at 3,800 × g and eluted chromatin supernatants were treated with proteinase K for 2 h at 65 °C and DNA was purified using a NucleoSpin gel and PCR clean-up kit from Macherey-Nagel.

**ChIP-seq.** Cells were cross-linked as above, but then cell pellets were resuspended in 1 ml of Extraction buffer (0.25 M sucrose, 10 mM tris-HCl pH 8.0, 10 mM MgCl₂, 1% Triton X-100, 1 mM DTT, PIC 1 × ) and incubated on ice for 10 min. Cells were spun at 3,000 × g at 4 °C and supernatant was discarded. Pellet was resuspended in 300 μl of Nuclei Lysis Buffer (50 mM TrisHCl pH 8.0, 10 mM

EDTA, 1% SDS, 1 × PIC). Suspension was transfered in Diagenode tubes and sonicated in a Bioruptor Pico (12 cycles). Chromatin was spun at 14,000 g for 10 min at 4 °C. Fragmentation was evaluated by running a fraction of the chromatin on a 1.2% Agarose gel. Chromatin was sonicated once again if necessary to obtain the appropriate range between 100 and 500 bp. For immunoprecipitation, chromatin was diluted 10-fold in ChIP dilution buffer (1.1% TritonX-100, 1.2 mM EDTA, 16.7 Tris-HCl pH 8.0, 167 mM NaCl, plus a protease inhibitor cocktail). Chromatin was precleared by adding 10–15 μl Protein A/G beads and incubated for 1 h at 4 °C. Samples were spun at 100 × g (4 °C) for 1 min. 6 μg of anti-H2A.J or 1.6 μg of anti-H2A (Abcam 18255) were added and each sample was incubated overnight at 4 °C. The next day, 30 μl of Agarose A/G beads were added and incubated during 4 hours at 4 °C. Immune complexes were recovered by spinning down samples at 100 g for 1 min. 50 μl was collected to generate input DNA. Beads were washed 3 times using low salt wash buffer (150 mM NaCl,0.1% SDS, 1% Triton X-100, 20 mM EDTA, 20 mM Tris-HCl pH 8.0), 1 time using High Salt Wash buffer (500 mM NaCl,0.1% SDS, 1% Triton X-100, 20 mM EDTA, 20 mM Tris-HCl pH 8.0), 1 time using LiCl buffer (0.25 M LiCl, 1% Igepal CA-630,1% Sodium deoxycholate, 1 mM EDTA, 10 mM Tris-HCl pH 8.0) and one time using TE Buffer (10 mM TrisHCl pH 8.0, 1 mM EDTA). Beads were resuspended in 50 μl of elution buffer (1% SDS, 0.1M NaHCO₃, 10 mM DTT) and incubated with 2 μl of RNase for 30 min at 37 °C. This was followed by a 2 h incubation with 2.5 μl of proteinase K and glycogen (0.4 mg − 1 ml final) at 37 °C. Reverse crosslinking was performed by incubating overnight at 65 °C. To purify the immunoprecipitated DNA, supernatants were collected in fresh tubes and incubated with 2.2 X SPRI beads for 4 min at RT and 4 min on a magnetic rack. Supernatants was discarded and beads were washed 2 times with 80% EtOH. Beads were air-dried for 4 min, beads were resuspended in 35 μl of TrisHCl pH 8.0 and incubated 4 min at RT and then 4 min on a magnetic rack. Supernatant containing purified DNA were collected. DNA was quantified using a Qbit fluorometer. Illumina ChIP-seq libraries were then prepared with the NEXTflex CHIP-seq kit and Barcodes from Bioo Scientific. 50 million single-end reads were acquired per sample. The fastq files were deposited at the SRA database under accession numbers SRR3824080, SRR3824087, SRR3824088 and SRR3824089.

**ChIP-seq data processing and analyses.** After quality filtering and adapter trimming using ea-utils fastq-mcf[54], 50 bp reads were mapped to the human genome (hg19) using BWA-MEM[55], considering only unique alignment. Optical and PCR duplicates were then removed using PicardTools (http://picard.sourceforge.net) and reads mapping in blacklisted region defined by the ENCODE consortium filtered using BedTools[56]. Tag density profiles were generated using deepTools bamCoverage[57] and a smoothing kernel with a 50-bp bandwidth. Cross-correlation was computed as described previously[31]. For each chromosome $c \in C$, the tag count vectors $n_c^+$ and $n_c^-$ were calculated to give the number of tags whose 5′ ends map to the position $x$ on the strand + or -. Strand cross-correlation for a strand shift δ was then calculated as

$$X(\delta) = \sum_{c \in C} \frac{N_c}{N} \times P\left[n_c^+\left(x + \frac{\delta}{2}\right), n_c^-\left(x - \frac{\delta}{2}\right)\right]$$

where $P[a,b]$ is the Pearson linear correlation coefficient between vectors a and b, $C$ is the set of all chromosomes, $N_c$ is the number of tags mapped to a chromosome $c$ and $N$ is the total number of tags. For the meta-profiles, read counts (RPM) in non-overlapping 60 bp windows around transcription start sites (TSS) were quantified for the H2A.J, H2A, and input DNA libraries and normalized to the total number of reads. The RPM values were then used to calculate log2(H2A/Input) and log2(H2A.J/Input) around the TSS.

**RT–qPCR.** RNA was extracted from 500,000 cells using Nucleospin RNA kits (Machery-Nagel). cDNA was prepared from 1 μg of RNA using random hexamer primers or oligo-dT as indicated. Quantitative PCR was performed with a Platinum SYBR Green qPCR SuperMix-UDG kits (Invitrogen) and an IQ5 thermocycler (Bio-Rad) following the manufacturer's protocols. Values were normalized to *GAPDH* RNA levels for human cells and to *GAPDH* and *PPIA* levels for mouse organs. DNA primers for the PCR are shown in Supplementary Table 3.

**Luminex assay of secreted inflammatory factors.** Conditioned media was isolated from four independent cultures of WI-38hTERT fibroblasts expressing either sh-NoTarget or sh3-*H2AFJ* RNAs. Media was analyzed from cells in proliferation or after inducing senescence by incubation with 20 μM etoposide for 9 days followed by 1 day in medium without etoposide. Cell numbers in the culture dishes were determined at the same time that media was collected and passed through 20 μM filters before freezing at − 80 °C until analysis. A panel of 63 human cytokines/chemokines/growth and adhesion factors were immunoassayed in parallel using Luminex technology. This assay was performed by the Human Immune Monitoring Center (HIMC) at Stanford University. Human 63-plex kits were purchased from eBiosciences/Affymetrix and used following the manufacturer's protocol with minor modifications. Beads were added to a 96 well plate and washed in a Biotek ELx405. Samples were then added to the plate containing the mixed antibody-linked beads and incubated at room temperature for 1 hour followed by overnight incubation at 4 °C with shaking at 500-600 r.p.m. Following the

overnight incubation plates were washed and then biotinylated detection antibody added for 75 min at room temperature with shaking. Plates was rewashed and streptavidin-PE was added. After incubation for 30 min and wash at room temperature, reading buffer was added to the wells and samples were measured with a Luminex 200 instrument with a lower bound of 50 beads per sample per cytokine. Custom assay control beads (Radix Biosolutions) were added to all wells. Fluorescent intensities were converted to concentrations using standard curves. The protein concentrations in the conditioned media were then normalized to the number of cells in the culture dishes at the time of collection.

**Data availability.** The Illumina idat files have been deposited in the GEO database under accession number GSE62701. RNA-seq data was deposited at the EBI Array Express under accession number E-MTAB-4920. The ChIP-seq fastq files were deposited at the SRA database under accession numbers SRR3824080, SRR3824087, SRR3824088 and SRR3824089.
R scripts used for the transcriptomic analyses are available on GitHub (https://github.com/CarlMann/sh-H2AFJ-transcriptome-analysis).

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

## Acknowledgements

C.M. received funding from the Fondation pour la Recherche Médicale (DEP20131128527), the Association pour la Recherche sur le Cancer, the Comité de l'Essonne de la Ligue Contre le Cancer, and the CEA Plasticity and Instability of the Genome Program. K.C. and C.C. were doctoral fellows in the CEA International PhD. Program (Irtelis). K.C. received prolongation doctoral funding from the ARC foundation. B.A.B. is supported by the Dean's fellowship at Stanford University and K99 AG049934 from NIA. C.E.R. received funding from the German Research Foundation (RU 821/3-1) and the European Atomic Energy Community's Seventh Framework Programme (FP7/2007-2011) under grant agreement n° 249689 (Low Dose Research towards Multi-disciplinary Integration). M.P.S. was funded by NIH grants 5P01GM09913005 and 5P50HG0773502. C.R. and W.M.B. were funded by the intramural program of the Center for Cancer Research, National Cancer Institute, National Institutes of Health. We thank Virginie Lavilla for help with qPCR experiments, the eBio platform (Institut Français de Bioinformatique) for bioinformatics support, and the Stanford Human Immune Monitoring Center for performing the Luminex immunoassay.

## Author contributions

K.C. and F.F. performed mass spectrometry experiments and analyses, C. Coudereau, C. Carvalho, C.E. Redon, W.M.B., R.C., K.C., C.M., and J.Y.T. characterized shRNA vectors and performed phenotypic analyses, M.C.N., H.V., C. Coudereau, C.M. and C.E. Redon prepared and characterized anti-H2A.J antibodies, C.E. Rübe. and N.S. performed IHC on mice and human skin samples, C. Coudereau, C.D., and R.O. performed Illumina Bead Chip experiments, K.C. and Z.M. prepared RNA-seq libraries, J-F.D., O.B., and M.P.S. provided crucial resources, C.M., B.A.B., K.C., C.D., R.O., C. Coudereau, J.Y.T., and P.F.R. performed bio-informatic analyses, C.W. and D.B. performed the mouse DMBA + TPA experiments, C. Coudereau and B.A.B. performed the ChIP experiments, K.C. and C.M. coordinated and oversaw the experiments, and C.M. wrote the manuscript with input from all authors.

## Additional information

**Competing interests:** The authors declare no competing financial interests.

