## [Peer Review File · Nature Communications]

Reviewers' Comments:

Reviewer #1 (Remarks to the Author)

The manuscript from Contrepois and colleagues uncovers interesting changes in the abundance of histone H2AJ in senescent cells. H2AJ is a minor variant that is poorly characterized in the literature. A more detailed characterization of the function of this variant would be a significant advance in the field and of wide interest. However, the current study is largely descriptive and does not provide any biologically relevant function for the changes observed in the abundance or localization of H2AJ. In addition, there are a number of issues with the experiments and data presented that are detailed below.

1. A major issue with this manuscript is that there is essentially no validation provided for the H2AJ-specific antibody. Given that H2AJ is different from canonical H2A by only 5 amino acids, it is critical for the authors to definitively demonstrate that their antibody recognizes H2AJ but does not recognize any other form of H2A. Without this validation, it is impossible to interpret much of the data in the manuscript.
2. The authors show that manipulating the levels of H2AJ alters various gene expression programs. For this result to be meaningful, the authors need to show that this is not just a non-specific effect of altering global H2A levels. The expression of other H2A variants should also be altered to test their effect on gene expression.

Reviewer #2 (Remarks to the Author)

Carl Mann and colleagues discovered that the histone variant H2A.J accumulates in senescent human fibroblasts where it plays a role in the senescent associated secretory phenotype. The data is very well presented and of high quality and the conclusions contribute to a better understanding of the senescence phenotype. Importantly, authors provide a much-needed new biomarker for senescent cells in vivo. Authors may consider discussing or clarifying the following points

- 1- H2A.J increases the expression of SASP genes but but accumulates also at promoters of genes that are not upregulated in senescence (Fig 6). How is that specificity achieved? Is H2AJ post-translational modified in senescent cells? Can they use their mass spectra data to predict post-translational modifications. Also, in the legend of Fig 6 it will be helpful to indicate the kind of senescence used for this experiment.
- 2- Punctate staining of H2A.J may suggest localization at PML bodies or DNA damage foci. Since PML and DNA damage regulate several traits associated to senescence it could be interesting to discuss this point or provide colocalization data with PML or 53BP1.
- 3- Punctate staining can be appreciated in fig 9 but it was not mentioned in the text.

Reviewer #3 (Remarks to the Author)

In this paper, Contrepois et al. present data arguing that H2A.J is a novel marker for senescence.

Several main points are made:

1. H2A.J accumulates in senescent fibroblasts with persistent DNA damage
2. H2A.J accumulation in senescent cells with persistent DNA damage involves a post-transcriptional component.
3. H2A.J overexpression increases inflammatory gene expression in proliferating cells
4. H2A.J is required for the normal production of SASP factors in senescence.
5. ChIP experiments indicate that H2A.J deposition is increased in senescent cells and that H2A.J is widely distributed, but depleted at transcription start sites.
6. H2A.J expression increases with age in mice in a tissue-specific fashion and may be a new biomarker of some senescent stem and skin cells.

Comments:

1. Authors show that H2A.J overexpression and KD affect the expression of a set of genes, and mentioned that H2A.J KD doesn't affect the entry and maintenance of senescence state.

-How does H2A.J overexpression affect senescence entry and maintenance?

-The data of induction of proliferating cells into senescent cells upon H2A.J depletion or overexpression is missing.

2. Authors said KD and overexpression of H2A.J didn't affect the efficiency of entry and maintenance of senescent cells. My understanding is that H2A.J accumulates in senescent cells; however, upon KD of H2A.J senescence is not affected.

-Can H2A.J really be considered as a marker for senescent cells?

3. Authors said KD and overexpression of H2A.J didn't affect the efficiency of entry and maintenance of senescent cells.

-Maybe low residual level of H2A.J is already enough for senescence entry and maintenance.

The authors should perform a KO of H2A.J in cells and test senescence induction.

4. By transcriptome analysis, authors discovered a set of genes that is regulated by H2A.J, however in ChIP analysis, H2A.J equally distributes to H2A.J dependent and independent genes.

-How to link this change in the transcriptome to the possible function of H2A.J in senescence state?

5. Authors write that 'H2AFJ mRNA was not expressed in higher levels in senescent cells compare to quiescent cells, but H2A.J protein levels are much higher in senescent cells'. There are several possibilities to explain this.

-What is the main reason for low protein level of H2A.J in proliferating cells or for its increased level in senescent cells? It would be useful to clarify this.

-How do expression and protein level of canonical H2A change in senescent state comparing to proliferating state?

NCOMMS-16-16310_H2A.J : Point by point response to reviewer's comments.

We thank Nature Communications and the reviewers for their interest in our work. Please find below our response to the comments of reviewers. In our revised manuscript, modified text is indicated in italics.

Reviewer #1 (Remarks to the Author):

The manuscript from Contrepois and colleagues uncovers interesting changes in the abundance of histone H2AJ in senescent cells. H2AJ is a minor variant that is poorly characterized in the literature. A more detailed characterization of the function of this variant would be a significant advance in the field and of wide interest. However, the current study is largely descriptive and does not provide any biologically relevant function for the changes observed in the abundance or localization of H2AJ. In addition, there are a number of issues with the experiments and data presented that are detailed below.

We show that the accumulation of H2A.J in senescence facilitates expression of inflammatory genes that are part of the SASP (Senescent-Associated Secretory Phenotype). It seems to us that this qualifies as a biologically relevant function for the observed increase in H2A.J abundance.

1. A major issue with this manuscript is that there is essentially no validation provided for the H2AJ-specific antibody. Given that H2AJ is different from canonical H2A by only 5 amino acids, it is critical for the authors to definitively demonstrate that their antibody recognizes H2AJ but does not recognize any other form of H2A. Without this validation, it is impossible to interpret much of the data in the manuscript.

We provided multiple independent data showing the specificity of our anti-H2A.J Ab :

-Fig. 1, Fig. 2, and old Fig. 8 (new Fig. 9) : the intensity of the H2A.J band in Western blots is in excellent agreement with the amount of H2A.J determined by mass spectrometry in multiple conditions, in several cell lines, and in several mouse organs.

- Supplementary Fig. 5a,b: 2 independent shRNAs were identified that are effective in knocking-down *H2AFJ* mRNA. The H2A.J band identified by our antibody in Western blots is greatly decreased in the cells expressing these shRNAs. Furthermore, when we ectopically express a Flag-HA-tagged cDNA encoding H2A.J, we observe a novel band that migrates at the expected size for the Flag-HA-H2A.J. These data are very strong indications of our Ab specificity in Western blots.

- Supplementary Fig. 7- cells expressing an shRNA to *H2AFJ* show reduced staining with our anti-H2A.J Ab when compared to sh-NoTarget cells

-old Supplementary Fig. 10 (new Supp. Fig. 11) : senescent cells show much higher immunofluorescent staining with our anti-H2A.J antibody compared to proliferating cells, in agreement with the abundance difference observed by mass spectrometry.

-old Supplementary Fig. 11 (new Supp. Fig. 12) : the anti-H2A.J immunofluorescence signal observed in formalin-fixed mouse tissue sections is completely eliminated by competition with a molar excess of the H2A.J C-terminal peptide used for immunization, but not with a C-terminal canonical H2A peptide.

Thus, we demonstrated the specificity of our anti-H2A.J antibody through 5 different criteria in our m/s. Furthermore, in the revised version of this m/s, we provide a Western blot showing that our antibody directed to the H2A.J C-terminus does not recognize an H2A.J mutant in which the specific C-terminus of H2A.J was exchanged for the C-terminus of canonical H2A type 1 (new Supplementary Fig. 9B).

In order to make explicit these multiple verifications of the specificity of our antibody, we have now added an antibody validation section to the Methods.

2. The authors show that manipulating the levels of H2AJ alters various gene expression programs. For this result to be meaningful, the authors need to show that this is not just a non-specific effect of altering global H2A levels. The expression of other H2A variants should also be altered to test their effect on gene expression.

We have responded to this request by testing the specificity of H2A.J by making a mutant in which we exchanged the C-terminal sequence specific to H2A.J with the C-terminal sequence from a canonical H2A. We find that this mutant is unable to complement the defect in inflammatory gene expression upon knock-down of endogenous H2A.J (Fig. 6 and Supp. Fig. 9). Thus, H2A.J has a specific role in promoting inflammatory gene expression that cannot be fulfilled by a non-specific H2A sequence. As we mentioned in the Introduction to our paper, the macroH2A.1 variant has already been shown to promote SASP gene expression in senescence (Chen et al., 2015). Thus, there are at least 2 different H2A variants that contribute to SASP gene expression. We do not feel that it is informative to alter the expression of yet other H2A variants in senescence because the results would have no bearing on our conclusions concerning a specific role for H2A.J in senescence.

Reviewer #2 (Remarks to the Author):

Carl Mann and colleagues discovered that the histone variant H2A.J accumulates in senescent human fibroblasts where it plays a role in the senescent associated secretory phenotype. The data is very well presented and of high quality and the conclusions contribute to a better understanding of the senescence phenotype. Importantly, authors provide a much-needed new biomarker for senescent cells in vivo. Authors may consider discussing or clarifying the following points

1- H2A.J increases the expression of SASP genes but but accumulates also at promoters of genes that are not upregulated in senescence (Fig 6). How is that specificity achieved? Is H2AJ post-translational modified in senescent cells? Can they use their mass spectra data to predict post-translational modifications. Also, in the legend of Fig 6 it will be helpful to indicate the kind of senescence used for this experiment.

We have been unable to detect post-translational modification of H2A.J in senescent cells by mass spectrometry. However, there are two technical caveats to this conclusion. First, our sensitivity is limited to about 1% modification of H2A.J molecules. If H2A.J modification was limited to the approximately 250 genes that it regulates, then we would roughly estimate that only approximately $250 \times 10 \text{ kb}$ (large estimate of promoter region) / $3 \times 10^6 \text{ kb}$ (human genome size) = 0.08% of H2A.J molecules would be modified, which is much below our detection sensitivity. A second limitation is that our profiling method would have detected modifications such as phosphorylation, acetylation, or methylation, but not more bulky modifications such as ubiquitinylation or sumoylation. Thus, we cannot rule out a role for post-translational modification of H2A.J in achieving specificity, but we have no direct evidence for such a mechanism either.

We have now indicated in the legend of Fig. 6 that the senescent state was induced by etoposide treatment. We thank the reviewer for pointing out this oversight.

2- Punctate staining of H2A.J may suggest localization at PML bodies or DNA damage foci. Since PML and DNA damage regulate several traits associated to senescence it could be interesting to discuss this point

or provide colocalization data with PML or 53BP1.

3- Punctate staining can be appreciated in fig 9 but it was not mentioned in the text.

The punctate H2A.J staining in Fig. 9 colocalizes with punctate regions of heterochromatin that are characteristic of mouse cells. In order to make this clearer, we are providing a new version of Fig. 9 containing a blow-up of hair follicle stem cell nuclei stained separately with Hoechst to visualize DNA and with anti-H2A.J. These images show that H2A.J staining is closely correlated with DNA density. This result is consistent with the ChIP-seq data indicating that H2A.J is widely deposited in senescent cell chromatin. Fig. 10b shows staining of human skin with anti-H2A.J and anti-53BP1 Abs. H2A.J was broadly distributed throughout the nuclei, which was in striking contrast to the small number and size of discrete 53BP1 foci.

Reviewer #3 (Remarks to the Author):

In this paper, Contrepois et al. present data arguing that H2A.J is a novel marker for senescence.

Several main points are made:

- 1. H2A.J accumulates in senescent fibroblasts with persistent DNA damage*
- 2. H2A.J accumulation in senescent cells with persistent DNA damage involves a post-transcriptional component.*
- 3. H2A.J overexpression increases inflammatory gene expression in proliferating cells*
- 4. H2A.J is required for the normal production of SASP factors in senescence.*
- 5. ChIP experiments indicate that H2A.J deposition is increased in senescent cells and that H2A.J is widely distributed, but depleted at transcription start sites.*
- 6. H2A.J expression increases with age in mice in a tissue-specific fashion and may be a new biomarker of some senescent stem and skin cells.*

Comments:

1. Authors show that H2A.J overexpression and KD affect the expression of a set of genes, and mentioned that H2A.J KD doesn't affect the entry and maintenance of senescence state.

-How does H2A.J overexpression affect senescence entry and maintenance?

-The data of induction of proliferating cells into senescent cells upon H2A.J depletion or overexpression is missing.

The data showing that depletion of H2A.J does not affect induction or maintenance of senescence was shown in Supplementary Fig. 6 of our paper. We have now added data showing that H2A.J overexpression does not affect senescent induction (new Supp. Fig. 6A,B).

2. Authors said KD and overexpression of H2A.J didn't affect the efficiency of entry and maintenance of senescent cells. My understanding is that H2A.J accumulates in senescent cells; however, upon KD of H2A.J senescence is not affected.

-Can H2A.J really be considered as a marker for senescent cells?

SASP expression is widely considered to be one marker of senescence, and we show that H2A.J accumulation occurs in senescence with persistent DNA damage and contributes to SASP expression. It thus appears to be a legitimate marker of some, but not all types of senescent cells. We note furthermore that the most often cited marker of senescent cells, SA- β -galactosidase activity, is not required for senescence (Lee et al., Aging Cell 2006. PMID :16626397).

3. *Authors said KD and overexpression of H2A.J didn't affect the efficiency of entry and maintenance of senescent cells.*

-Maybe low residual level of H2A.J is already enough for senescence entry and maintenance.

The authors should perform a KO of H2A.J in cells and test senescence induction

We recently obtained a homozygous knock-out mouse for H2A.J. It is viable and fertile. The initial phenotypic characterization of this mouse will take a minimum of 1-2 years (even longer for the aging studies). However, we have derived H2A.J knock-out MEFs and they undergo senescence in a similar fashion to WT MEFs (please see the Reviewer Figure attached to the end of this point by point response to reviewers). Thus, our conclusion from shRNA knock-down studies with human fibroblasts that H2A.J is not required for induction of senescence is true as well for these MEF knock-out cells. We would prefer not including this data in our paper because it does not add any interesting new information to our characterization in human fibroblasts, but would require us to provide significant supplementary documentation concerning the construction and the verification of the knock-out mouse. We would prefer presenting this characterization of the knock-out cells in the proper context for which we made them : in analyzing the role of H2A.J in tissue function and aging in the mouse. Nevertheless, if this reviewer and Nature Communications feel that it is essential to include this characterization of the knock-out MEFs, we will provide it as an additional Supplementary figure.

4. *By transcriptome analysis, authors discovered a set of genes that is regulated by H2A.J, however in ChIP analysis, H2A.J equally distributes to H2A.J dependent and independent genes.*

-How to link this change in the transcriptome to the possible function of H2A.J in senescence state?

If I understand the question correctly, this reviewer is asking how H2A.J facilitates the expression of inflammatory genes in senescent fibroblasts ? We are certainly interested in this question, but we have not yet determined the mechanism. As we stated in the Discussion : “ The specificity of H2A.J action thus cannot be due to its preferential deposition at inflammatory genes. This result suggests that other factors must act in concert with H2A.J to promote strong inflammatory gene expression. The distinct C-termini of H2A variants contribute to their functional specificities (Bönisch and Hake, 2012). The unique H2A.J N and C-termini may affect chromatin compaction and interact differentially with protein partners. It is also possible that H2A.J accumulation in senescence may act indirectly by displacing other H2A variants.” The new data that we provide in Fig. 6 suggests that the C-terminus of H2A.J is indeed important for its function in stimulating inflammatory gene expression in senescence, but we don't know how. The above speculations are all reasonable and we are continuing to work on this question. We would like to point out that defining the mechanistic basis of histone variant function is not trivial. There have been hundreds of articles on the H2A.Z variant and its mechanisms of action are only partly understood. We provide the first description of H2A.J at the protein level with a demonstration of its function in promoting inflammatory gene expression in senescence, and we hope that this will suffice for an initial publication.

5. *Authors write that 'H2AFJ mRNA was not expressed in higher levels in senescent cells compare to quiescent cells, but H2A.J protein levels are much higher in senescent cells'. There are several possibilities to explain this.*

-What is the main reason for low protein level of H2A.J in proliferating cells or for its increased level in senescent cells? It would be useful to clarify this.

In our paper, we provide data showing that mRNA levels do not suffice to explain the differential accumulation of H2A.J protein in senescent fibroblasts compared to serum-starved fibroblasts. We have not observed differences in H2A.J stability in proliferating versus senescent cells after blocking neo-synthesis with cycloheximide. By elimination, this negative result suggests that the translatability of the *H2AFJ* mRNA may be facilitated in senescent cells. However, it has been difficult to demonstrate this directly. H2A.J levels accumulate slowly over 2 weeks in senescent cells. A small increase in translation efficiency over 2 weeks might suffice to explain the increased accumulation of H2A.J in senescence. We tried transferring the H2AFJ 3'-UTR to a GFP reporter construct, but this did not suffice to confer differential accumulation of the GFP reporter in senescent cells. Further work will be required to understand how H2A.J accumulation is differentially increased in senescent cells.

-How do expression and protein level of canonical H2A change in senescent state comparing to proliferating state?

This information was provided in Figs. 1 and 3 of our paper. We noted (p.3): “Our histone profiling also revealed changes in the levels of some canonical H2A species. The proportion of canonical H2A-type 1C (Uniprot Q93077) increased in all non-proliferative states (quiescence and senescence), whereas other canonical histones, such as the H2A-type 1 (Uniprot P0C0S8) and H2A-type 1B/E (Uniprot P04908), decreased in senescent cells with persistent DNA damage. However, H2A.J was the only H2A species that was present at low levels in proliferating cells and high levels in senescent cells with DNA damage.”

On p. 5 : “With the exception of *HIST1H2AC* mRNA, most of the canonical H2A mRNAs dropped dramatically in non-proliferating cells (**Fig. 3a**), as expected from their known transcriptional and post-transcriptional regulation⁹. The persistence of *HIST1H2AC* mRNA in non-proliferating cells may be explained by the presence of a polyadenylation site downstream of its stem-loop motif. Polyadenylated *HIST1H2AC* mRNA indeed accumulated in quiescent and senescent fibroblasts (**Fig. 3b**) and likely contributes to the accumulation of H2A-type 1C protein in these states (**Fig. 1b**).”

Reviewer Figure. H2AFJ-ko MEFs can be induced into senescence similarly to WT MEFs. (A) a TALEN-mediated 7 bp deletion at the beginning of the *H2AFJ* coding sequence creates a frame shift with a premature stop codon that is predicted to yield a null phenotype. Shown is the first half of the coding sequence from the ATG start codon for the WT and H2AFJ-Δ7 sequence. The red box shows the position of the 7 bp deletion within the WT sequence. The H2AFJ-Δ7 sequence shows the frameshift and premature stop codon position. (B) Immunofluorescence with anti-H2A.J Abs shows expression of H2A.J in congenic C57Bl/6 WT MEFs, but not H2AFJ-Δ7 ko MEFs. (C) Top-down mass spectrometry of H2A in the kidneys of 3 month old WT and homozygous mutant H2AFJ-Δ7 mice showing the absence of H2A.J in the ko mouse. (D) WT and H2AFJ-Δ7 ko MEFs were treated with 2.5 μM etoposide for 1 week. Proliferating cells or etoposide-treated cells were incubated with BrdU for 24 hours. Similar inhibition of DNA synthesis was observed for both cells after etoposide treatment. (E) SA-β-gal staining shows similar levels of senescence induction for WT and H2AFJ-Δ7 ko MEFs after treatment with 2.5 μM etoposide for 1 week.

Reviewers' Comments:

Reviewer #1 (Remarks to the Author)

The revised manuscript from Contrepois and colleagues does not address any of the issues raised in my initial review. First, while altering the expression of H2A.J changes the expression of a small number of genes, these alterations have no effect on any detectable biological phenotype. Hence, this H2A variant really hasn't been linked to any real biological process. Second, while the authors describe numerous reasons that use to justify the specificity of their antibody, most of these are based on the somewhat circular logic that the antibody must be specific because it behaves in the way they want it to behave. There are two definitive ways that the specificity of the antibody can be demonstrated. Western blots could be performed with recombinant forms of H2A.J and canonical H2A. Alternatively, Western blots could be performed with histones isolated from WT mouse cells and from cells where the gene for H2A.J has been deleted. As the authors state in their response, they have made H2A.J KO mice and so this experiment should be trivial to perform. The third issue raised was about the specificity of the KD of H2A.J with respect to alterations in gene expression. The authors disregard this issue based on published literature showing that macroH2A also alters SASP gene expression. The analysis of macroH2A is not really relevant as macroH2A is a replication-independent H2A variant and H2A.J is a replication-dependent variant. The replication-dependent and -independent histone variants are regulated in very different ways and it is likely that they function in different ways, as well.

Reviewer #2 (Remarks to the Author)

The authors has answered satisfactorily to my questions and those of other reviewers. The manuscript reports a novel biomarker for senescence and identifies a new factor regulating SASP gene expression. The paper will stimulate research into the role of H2AJ on cytokine gen expression both in senescent and cancer cells.

Reviewer #3 (Remarks to the Author)

The authors have adequately answered my initial requests and I am now satisfied with the manuscript.

Reviewer #1 (Remarks to the Author):

The revised manuscript from Contrepois and colleagues does not address any of the issues raised in my initial review. First, while altering the expression of H2A.J changes the expression of a small number of genes, these alterations have no effect on any detectable biological phenotype. Hence, this H2A variant really hasn't been linked to any real biological process.

We respectfully submit that our demonstration of a role for H2A.J in facilitating the expression of inflammatory genes in senescence represents a real biological process. The Luminex assay (Fig. 5f and Supp. Fig. 8) showed strong inhibition in the levels of several chemokines/cytokines in the conditioned media of senescent cells after knock-down of H2A.J.

Second, while the authors describe numerous reasons that use to justify the specificity of their antibody, most of these are based on the somewhat circular logic that the antibody must be specific because it behaves in the way they want it to behave. There are two definitive ways that the specificity of the antibody can be demonstrated. Western blots could be performed with recombinant forms of H2A.J and canonical H2A. Alternatively, Western blots could be performed with histones isolated from WT mouse cells and from cells where the gene for H2A.J has been deleted. As the authors state in their response, they have made H2A.J KO mice and so this experiment should be trivial to perform.

We indeed wanted to demonstrate that our antibody, raised against a discriminating C-terminal epitope, specifically recognizes H2A.J. To test this, we defined a set of objective criteria for specificity: in situations in which we had data based on RT-qPCR and mass spectrometry for decreases or increases in H2A.J levels, our affinity-purified antibody should accurately reflect those differences in both Western blots and immunofluorescence.

- We validated the knock-down efficiency of 2 different sh-H2AFJ sequences and verified that our anti-H2A.J Ab showed reduced signals in cells expressing these sh-RNAs compared to sh-NoTarget RNA by both Western blotting and immunofluorescence.
- Mass spectrometry indicated strikingly different levels of H2A.J in mouse brain, liver, and kidney that were faithfully correlated with the Western blot signals using our anti-H2A.J antibody.
- When we over-expressed a tagged version of H2A.J, we observed a novel band in Western blots at the expected size of the tagged protein.
- The immunofluorescent signal observed in tissue sections was competed with a peptide from the H2A.J C-terminus, but not the C-terminus of canonical H2A.
- Furthermore, in our revised manuscript, we provided additional data in the form of a Western blot (new Supplementary Figure 9B) showing that our Ab does NOT recognize a form of H2A.J in which the specific C-terminus of H2A.J has been replaced with the C-terminus of canonical H2A. Our Ab was raised to the specific C-terminus of H2A.J. This Western blot showed that our Ab recognized H2A.J, but not a mutant of H2A.J containing the C-terminus of canonical H2A.

As our Ab was raised against the specific C-terminus of H2AJ, this last experiment is the equivalent of a Western blot performed with recombinant forms of H2A.J and canonical H2A and corresponds to one of the two criteria requested by Reviewer 1. We inserted Supp. Fig. 9 at the end of this response for easy reference.

We are also hereby providing the second demonstration requested by Reviewer 1: a Western blot of whole cell extracts from WT and H2A.J-ko MEFs, and of histones extracted from WT and

H2A.J-ko mouse kidney and liver, showing a strong specific signal for H2AJ. We are providing this data as Reviewer Fig. 2 at the end of this letter. Reviewer Fig. 1, presented in our first Response to Reviewers, described the H2A.J-ko mouse with the demonstration that the absence of H2A.J does not inhibit senescence of MEFs. Reviewer Fig. 2 also shows that our anti-H2A.J Ab detects a faint band in extracts from the H2A.J-ko MEFs (10% of the intensity of the band in extracts from WT MEFs) that is even fainter in the histones from the ko organs (less than 2% the intensity of the band in WT organs). Since this band migrates at the level of H2A/H2A.J, it may represent a weak cross-reaction with H2A. Mass-spectrometry analyses show that WT MEFs contain about 1 molecule of H2A.J for 100 molecules of H2A (Fig. 9a in our paper), and adult mouse liver/kidney contain about 10 molecules of H2A.J for 100 molecules of H2A (Fig. 9d). Since the cross-reacting band in ko-MEFs is 1/10 the level of the specific signal in WT-MEFs, we can calculate that our anti-H2A.J antibody has an affinity about 1,000-fold greater for H2A.J compared to canonical H2A. This value is in good agreement with quantitative Elisa peptide competition assays that we performed when we first prepared the antibody. The C-terminal H2A.J peptide competes the Elisa signal with a 1,000-fold higher affinity compared to the C-terminal canonical H2A sequence. We are providing this data as Reviewer Fig. 3.

In conclusion, the Western blot from H2A.J-ko cells demonstrates the high specific signal of our anti-H2A.J antibody in a manner consistent with the 6 other criteria above that we presented in our revised m/s to demonstrate its specificity. A further example of this specificity can be seen in the immunofluorescent image of H2A.J-ko MEFs versus WT MEFs (Reviewer Fig. 1b) in which our anti-H2A.J Ab unambiguously distinguishes the WT from the knock-out MEFs. We thank Reviewer 1 for his suggestions that allowed us to provide a more quantitative demonstration of the specificity of our antibody.

We would like to emphasize that this weak level of cross-reactivity (10% the level of the H2A.J-specific signal when H2A.J is expressed at low levels as in proliferating MEFs, and less than 2% the level of the H2A.J-specific signal when H2A.J is expressed at high levels as in adult mouse liver and kidneys) would not be expected to significantly contribute to the antibody signals in our data. As we detailed in our initial response to Reviewers, these results were further supported by orthogonal data (mass spectrometry and RT-qPCR) and additional controls (competition by an H2A.J peptide, but not a canonical H2A peptide).

The third issue raised was about the specificity of the KD of H2A.J with respect to alterations in gene expression. The authors disregard this issue based on published literature showing that macroH2A also alters SASP gene expression. The analysis of macroH2A is not really relevant as macroH2A is a replication-independent H2A variant and H2A.J is a replication-dependent variant. The replication-dependent and –independent histone variants are regulated in very different ways and it is likely that they function in different ways, as well.

H2A.J is a replication-independent H2A variant like macro-H2A. We demonstrated this in our paper (Fig. 3), and it was also shown in the only other article that has been published on the gene encoding H2A.J (Nishida, H., Suzuki, T., Tomaru, Y. & Hayashizaki, Y. A novel replication-independent histone H2a gene in mouse. BMC Genet. 6, 10 (2005)). The official name of the gene encoding H2A.J is *H2AFJ*. In the official histone gene nomenclature, the F is added to histone genes that are expressed in a replication-independent fashion. We noted in our m/s that H2A.J is clearly a replication-independent H2A variant.

Reviewer 1 wanted us to address the specificity of H2A.J knock-down with respect to alterations in gene expression. In response, we provided important additional data in our revised m/s showing that a mutant of H2A.J containing the C-terminus of canonical H2A in place of the H2A.J C-terminus cannot functionally complement the H2A.J knock-down (new Fig. 6 and new Supp. Fig. 9).

This result demonstrates specificity in H2A.J function. As we pointed out in response to this reviewer, knock-down of the macroH2A.1 variant has already been shown to affect SASP gene expression in senescence (Chen et al., 2015). Thus, there are at least 2 different H2A variants that contribute to SASP gene expression. We could knock-down a canonical H2A gene sequence and test its effect on gene expression, but no matter what the result, we do not understand what that could tell us about the function of H2A.J. What conclusion could we draw from such an experiment whether or not it affected SASP gene expression ?

Supplementary Figure 9. Expression levels of endogenous and ectopically expressed *H2AFJ* RNAs and *H2A.J* proteins in functional complementation experiments.

WI-38/hTERT cells were infected with pGIPz lentiviruses expressing a No Target shRNA (NT) or an sh3-*H2AFJ* RNA (sh). The sh3-*H2AFJ* cell line was super-infected with pTRIPz lentiviruses expressing sh3-resistant *H2AFJ* cDNAs encoding WT-*H2A.J* or a mutant *H2A.J* in which the C-terminus of *H2A.J* was substituted with the C-terminus of canonical *H2A* type 1 (*H2A.J*-Cter-*H2A*). Both the WT and mutant *H2A.J* contained an N-terminal Flag-HA tag. RNA was isolated from cells induced into senescence by etoposide treatment and compared to levels in proliferating sh-No Target cells. **(a)** RT-qPCR quantification of endogenous *H2AFJ* mRNA (dark grey) and total *H2AFJ* mRNA (endogenous + ectopically-expressed cDNAs encoding N-terminally-tagged Flag-HA-*H2A.J* WT and *H2A.J*-Cter-*H2A* proteins). Results show the mean and s.d. from 3 biological replicates. **(b)** Western blots of total cell extracts from the indicated cells with anti-*H2A.J*, anti-HA (hemagglutinin) tag, and anti-GAPDH antibodies. Note that the anti-*H2A.J* Ab directed to the specific C-terminus of *H2A.J* recognizes the Flag-HA-*H2A.J* protein but does not recognize the Flag-HA-*H2A.J*-Cter-*H2A* protein, whereas both proteins are recognized by the anti-HA tag Ab. Both the RT-qPCR and Western analyses indicate that the Flag-HA-*H2A.J*-Cter-*H2A* was mutant was expressed at slightly higher levels than the Flag-HA-*H2A.J* WT protein.

Reviewer Fig. 1. H2AFJ-ko MEFs can be induced into senescence similarly to WT MEFs. (A) a TALEN-mediated 7 bp deletion at the beginning of the *H2AFJ* coding sequence creates a frame shift with a premature stop codon that is predicted to yield a null phenotype (knock-out). Shown is the first half of the coding sequence from the ATG start codon for the WT and H2AFJ-Δ7 sequence. The red box shows the position of the 7 bp deletion within the WT sequence. The H2AFJ-Δ7 sequence shows the frameshift and premature stop codon position. **(B)** Immunofluorescence with anti-H2A.J Abs shows expression of H2A.J in congenic C57Bl/6 WT MEFs, but not H2AFJ-Δ7 ko MEFs. **(C)** Top-down mass spectrometry of H2A in the kidneys of 3 month old WT and homozygous mutant H2AFJ-Δ7 mice showing the absence of H2A.J in the ko mouse. **(D)** WT and H2AFJ-Δ7 ko MEFs were treated with 2.5 μM etoposide for 1 week. Proliferating cells or etoposide-treated cells were incubated with BrdU for 24 hours. Similar inhibition of DNA synthesis was observed for both cells after etoposide treatment. **(E)** SA-β-gal staining shows similar levels of senescence induction for WT and H2AFJ-Δ7 ko MEFs after treatment with 2.5 μM etoposide for 1 week.

Reviewer Fig. 2. Western blot of wild-type and *H2AFJ*-knock-out mouse cells demonstrates specificity of the anti-H2A.J antibody. Mouse embryonic fibroblasts (MEF) were prepared from 2 different WT embryos and 3 different *H2AFJ* Δ 7 knock-out embryos (see Reviewer Fig. 1 for a description of the *H2AFJ* Δ 7 mutation). Total cellular proteins were extracted from the primary MEFs within 3 passages. Histones were also extracted with acid from 6 month-old WT and *H2AFJ* Δ 7 liver and kidneys. Proteins were then analyzed by SDS-PAGE (15% polyacrylamide) and transferred to nitrocellulose membranes for immunoblotting. (A) LiCor Odyssey scan of the membrane after incubation with a 1/2000 dilution of rabbit anti-H2A.J antibody for 1 hour followed by anti-rabbit-800 secondary antibodies. (B) The same membrane was then incubated for 1 hour with rabbit anti-H3 antibody (Abcam 1791) followed by anti-rabbit-800 secondary antibodies and a second scan of the membrane. The grey scale signals for the H2A.J and H3 bands were quantified with ImageJ and the normalized H2A.J/H3 signal ratios are shown between panels A and B. (C) Coomassie staining of 15% gel containing the same samples.

Reviewer Fig. 3. Elisa peptide competition assay demonstrating a 1,000-fold higher affinity of the anti-H2A.J antibody for the C-terminal H2A.J peptide compared to a C-terminal canonical H2A peptide. Microplate wells containing the anti-H2A.J Ab were incubated with Acetylcholinesterase (ACE) conjugated to the C-terminal H2A.J peptide in the presence of competing peptides at the indicated concentrations. The binding of ACE-H2A.J to the wells was quantified colorimetrically with Ellman's reagent. The C-terminal H2A.J peptide (TESQKTKSK)(peptide-1 in the figure) binds the anti-H2A.J Ab with a 1,000-fold higher affinity than the C-terminal canonical H2A peptide (TESHHKAKGK) (peptide-3). This assay also showed that phosphorylation of the H2A.J peptide (TE-phospho-SQKTKSK)(peptide-2) decreased its affinity 100-fold.

Reviewers' Comments:

Reviewer #1 (Remarks to the Author)

The new data showing Western blots from H2AJ WT and KO mice is compelling and I am convinced that the antibodies are reasonably specific.

I would like to apologize for my error regarding my third issue. I mistakenly thought that H2AJ was referring to the H2AJ gene in histone cluster 1. Hence, my incorrect notion that H2AJ was a relocation-dependent histone.